# ADADAGRAD: Adaptive Batch Size Schemes for Adaptive Gradient Methods

Author One*          Author Two          Author Three
Affiliation One      Affiliation Two     Affiliation Three

**Abstract**

The choice of batch size in minibatch stochastic gradient optimization is critical for both optimization and generalization performance in large-scale model training. Although large-batch training is arguably the dominant paradigm in large-scale deep learning because of hardware advances, model generalization often deteriorates relative to small-batch training, leading to the so-called "generalization gap." To mitigate this issue, we investigate adaptive batch size strategies derived from adaptive sampling methods, which were originally developed for stochastic gradient descent. Given the strong interplay between learning rates and batch sizes, together with the prevalence of adaptive gradient methods in deep learning, we emphasize the need for adaptive batch size strategies in these settings. We introduce ADADAGRAD and its scalar variant ADADAGRAD-NORM, which progressively increase batch sizes during training while performing updates with ADAGRAD and ADAGRAD-NORM, respectively. We prove that ADADAGRAD-NORM converges with high probability at a rate of $\mathcal{O}(1/K)$ to a first-order stationary point of a smooth nonconvex function within $K$ iterations. ADADAGRAD also exhibits similar convergence properties when combined with a novel coordinate-wise variant of our adaptive batch size strategy. We corroborate our theoretical claims with image-classification experiments that highlight the merits of the proposed schemes in terms of both training efficiency and model generalization. Our work highlights the potential of adaptive batch size strategies for adaptive gradient optimizers in large-scale model training.

**Keywords:** adaptive batch size schemes, adaptive gradient methods, stochastic gradient methods, large-batch training, generalization gap

**Mathematics Subject Classification (2020):** 90C15, 90C30

## 1 Introduction

Large-scale optimization algorithms (Bottou et al., 2018) form the foundation of deep learning success in the era of generative AI. Minibatch stochastic gradient descent (SGD) (Robbins and Monro, 1951) and its many variants, together with batch-sampling techniques, are the main workhorses for training deep neural networks. However, training deep neural networks, such as those used in transformers, is notoriously challenging because of their high dimensionality and nonconvex loss landscapes. This complexity necessitates extensive hyperparameter tuning

---

*Corresponding author: author.one@email.com

and sophisticated training strategies to avoid premature divergence and training instabilities. Consequently, training deep learning models often appears more as an art than a science. The most critical hyperparameter is arguably the learning rate (or step size). Adaptive gradient methods with adaptive learning rates, such as ADADELTA (Zeiler, 2012), ADAGRAD (Duchi et al., 2011), and ADAM (Kingma and Ba, 2015), are now prevalent because they reduce the need for meticulous tuning and complex learning-rate schedules, which are typically required for SGD. Another important yet frequently overlooked hyperparameter is the batch size. It governs the trade-off between computational efficiency and model generalization by controlling the magnitude of noise in batch gradients. However, batch-size selection in deep learning remains largely heuristic, such as using a constant batch size for convolutional networks or a linear warmup for large language models (Brown et al., 2020; Rae et al., 2021; Hoffmann et al., 2022), and is usually predetermined before training begins. Furthermore, from a hardware-utilization perspective, using a large number of distributed computational resources (i.e., GPUs or TPUs) often necessitates large-batch training when parallel minibatch SGD (Zinkevich et al., 2010; Dean et al., 2012) is employed. In addition, the intricate relationship between learning rates and batch sizes deserves attention. Specifically, Smith et al. (2018) showed an equivalence between reducing step sizes and increasing batch sizes, but this principle applies mainly to SGD. The impact of varying batch sizes in adaptive gradient methods has not yet been fully explored.

In light of this, our work seeks to clarify how to determine suitable batch sizes for adaptive gradient methods. We aim to introduce automated schedules that dynamically decide when to increase the batch size, and by how much, based on training needs. Our approach is theoretically grounded and relies on the statistics of batch gradients at the iterates, thereby adapting to training dynamics. The proposed schedules can reduce the generalization gap in large-batch training while still making use of large batches in later stages of training. Our strategies are based on adaptive sampling methods (Byrd et al., 2012; Bollapragada et al., 2018a). In the context of deep learning, De et al. (2016, 2017) numerically demonstrated the effectiveness of these methods when combined with ADADELTA. However, the convergence properties of such adaptive sampling methods for adaptive gradient methods have not been thoroughly investigated, leaving a gap between theory and practice. Moreover, in existing adaptive sampling methods developed mainly for SGD, step sizes are often fixed or adjusted using backtracking line-search procedures (mainly for convex problems). This becomes computationally impractical or inefficient for large models, especially given the nonconvex nature of deep neural networks.

The development of adaptive batch size strategies for deep learning is not new; examples include Big Batch SGD (De et al., 2016, 2017), CABS (Balles et al., 2017), AdaBatch (Devarakonda et al., 2017), SimiGrad (Qin et al., 2021), and AdaScale SGD (Johnson et al., 2020), which adapts learning rates for large batches rather than directly adjusting batch sizes. However, these methods often lack a principled basis with rigorous convergence guarantees, or they are limited to analyses of SGD under restrictive conditions (e.g., convexity or the Polyak–Łojasiewicz condition), despite the prevalence of adaptive gradient methods in deep learning. Moreover, these approaches still require choices about learning-rate adjustment, such as line-search routines or schedulers, leaving a gap in full adaptivity. In addition, strategies for determining new batch sizes may rely on heuristic rules, such as geometric growth or decay rates (Qin et al., 2021).

## 1.1 Contributions

In this work, our objective is to bridge the gap between theory and practice for adaptive batch size schemes used with adaptive gradient methods. We introduce two main adaptive batch size schemes, grounded in the so-called *adaptive sampling methods* (Byrd et al., 2012; Friedlander and Schmidt, 2012; Bollapragada et al., 2018a), tailored to adaptive gradient methods. Our focus is on ADAGRAD and its norm variant ADAGRAD-NORM, which are among the simplest and most extensively studied adaptive gradient methods. Developing adaptive batch size schemes for these methods is therefore of both theoretical and practical interest.

The technical contributions of this work are threefold. From a theoretical perspective, we establish a sublinear convergence rate (with high probability) for our proposed methods when combined with ADAGRAD-NORM and ADAGRAD on smooth nonconvex objectives. This substantially broadens the existing body of work, which has mainly focused on SGD. Moreover, we relax the Lipschitz smoothness condition on the objective function by adopting the generalized smoothness concept introduced in Zhang et al. (2020b,a). This adaptation allows for more general and realistic applications in contemporary deep learning practice. On the empirical side, we demonstrate the effectiveness of our proposed methods through a range of numerical experiments on image-classification tasks. These experiments highlight the benefits of the adaptivity of our schemes, driven by both adaptive batch sizes and adaptive step sizes. Finally, we provide an efficient implementation of the proposed approach in PyTorch, using the `torch.func` module for efficient parallel computation of per-sample gradients via the vectorized map function `vmap`. To the best of our knowledge, our proposed methods are the first adaptive batch size schemes based on adaptive sampling methods for adaptive gradient methods that come with convergence guarantees, strong empirical performance, and efficient implementations in deep learning libraries.

**Theory–implementation distinction.** The convergence results in this paper are stated for an idealized exact-test version of the proposed adaptive batch-size schemes. In this version, the batch size at iteration $k$ is assumed to be chosen so that the corresponding population-level variance condition holds at $x_k$. The practical implementation, by contrast, uses empirical batch statistics computed from the current minibatch to select the next batch size. Thus, the theory establishes convergence for the population-level adaptive-sampling principle underlying the method, while the experiments evaluate a computationally feasible approximate implementation. We make this distinction explicit throughout the paper.

## 2 Related Work

We provide an overview of related work on batch sizes in model training and adaptive sampling methods, as well as on the convergence of stochastic gradient methods.

### 2.1 Large-Batch Training

In stochastic gradient optimizers, batch sizes play a crucial role in controlling the variance of stochastic gradients as estimators of full deterministic gradients. Although the noise induced

by stochastic gradients can be beneficial in nonconvex optimization, the trend toward larger batches in large-scale model training has become standard thanks to advances in parallel hardware such as GPUs, which substantially reduce training time. The notion of large-batch training in deep learning was popularized in Smith and Le (2018); Smith et al. (2018) and has been widely adopted in applications such as ImageNet classification (Goyal et al., 2017) and BERT training (You et al., 2020). Since LeCun et al. (2002), it has been recognized that model generalization can deteriorate in large-batch training. Large-batch training tends to yield loss landscapes with many sharp minima, which are harder to escape during optimization and hence can lead to worse generalization (Keskar et al., 2017). The impact of batch sizes has been examined more systematically in later work. For example, McCandlish et al. (2018) introduced an empirical model for large-batch training without rigorous theoretical proof, postulating the existence of critical batch sizes through extensive numerical simulations on convolutional neural networks (CNNs), LSTMs, and VAEs. Meanwhile, Kaplan et al. (2020) focused on transformers. Zhang et al. (2019) explored how critical batch sizes vary with optimizer characteristics, including momentum, preconditioning, and exponential moving averages, through both large-scale experiments and a noisy quadratic model. Granziol et al. (2022) applied random matrix theory to examine the batch Hessian, theoretically deriving learning-rate scaling rules as a function of batch size for both SGD (*linear*) and adaptive gradient methods (*square-root*), with experimental validation. Although the "generalization gap" can be narrowed by using larger learning rates proportional to batch size to maintain the gradient-noise scale (Hoffer et al., 2017; Smith and Le, 2018; Smith et al., 2018), it cannot be completely eliminated. However, Shallue et al. (2019) investigated the impact of batch sizes in the context of data parallelism and empirically characterized the effects of large-batch training, finding no evidence of degraded generalization performance.

## 2.2 Adaptive Sampling Methods

In stochastic optimization, a more theoretically grounded approach known as *adaptive sampling methods* has been developed for batch (or minibatch) algorithms. Byrd et al. (2012) introduced a method called the norm test, which adaptively increases the batch size throughout the optimization process. The rationale behind the norm test traces back to Carter (1991) and represents a more general condition than the one proposed by Friedlander and Schmidt (2012). Both approaches establish linear convergence when the batch size increases geometrically. Bollapragada et al. (2018a) proposed the augmented inner product test, which allows for more gradual increases in batch size than the norm test. Furthermore, Cartis and Scheinberg (2018) introduced a relaxation of the norm test that allows its condition to be violated with probability less than 0.5. This family of adaptive sampling methods belongs to the class of variance-reduced optimization algorithms (Johnson and Zhang, 2013; Reddi et al., 2016) widely used in machine learning (Gower et al., 2020).

Adaptive sampling methods have also been extended to problems beyond unconstrained optimization. Xie et al. (2023) proposed proximal extensions of the norm test and the inner product test for minimizing a convex composite objective consisting of a stochastic function and a deterministic, potentially nonsmooth function. Beiser et al. (2023) studied deterministic constrained problems, including cases with nonconvex objectives. Moreover, adaptive sampling

methods have been applied to a range of optimization problems and algorithms, including Bollapragada et al. (2018b) for L-BFGS, Berahas et al. (2022) for sequential quadratic programming (SQP) in equality-constrained problems, Bollapragada et al. (2023) for augmented Lagrangian methods, and Bollapragada and Wild (2023) for quasi-Newton methods.

## 2.3 Convergence Results of Stochastic Gradient Methods

We provide a brief overview of convergence results for SGD and adaptive gradient methods. The convergence of SGD for smooth nonconvex functions was first analyzed in Ghadimi and Lan (2013), under the assumption of a uniform bound on the variance of stochastic gradients. Arjevani et al. (2023) later established a tight lower bound. Bottou et al. (2018) extended the convergence result using the so-called *affine variance* noise model. Different assumptions on the second moment of stochastic gradients appearing in the literature were reviewed in Khaled and Richtárik (2023), with the goal of developing a general convergence theory for SGD on smooth nonconvex functions.

Another line of work concerns convergence guarantees in expectation for adaptive gradient methods. Ward et al. (2019, 2020); Li and Orabona (2019); Faw et al. (2022) each established a convergence rate of $\widetilde{\mathbb{O}}(1/\sqrt{K})$ for ADAGRAD under different assumptions on stochastic gradients. Défossez et al. (2022) gave a simple proof of the convergence of both ADAGRAD and a simplified version of ADAM through a unified formulation.

Regarding high-probability convergence bounds, Ghadimi and Lan (2013) established high-probability convergence for SGD with a properly tuned learning rate, assuming known smoothness and sub-Gaussian stochastic-gradient noise bounds. Under the same assumptions, Zhou et al. (2024); Li and Orabona (2020) obtained similar results for (delayed) ADAGRAD. Again, under different assumptions on stochastic gradients and using different proof techniques, Kavis et al. (2022); Faw et al. (2023); Wang et al. (2023); Attia and Koren (2023); Liu et al. (2023) derived high-probability convergence rates for ADAGRAD(-NORM).

# 3 Problem Formulation

In this section, we lay out the general problem formulation considered in this work.

## 3.1 Notation

We define $[\![n]\!] \coloneqq \{1, \dots, n\}$ for $n \in \mathbb{N}^* \coloneqq \mathbb{N} \setminus \{0\}$, $\mathbb{R}_+ \coloneqq [0, \infty)$, and $\mathbb{R}_{++} \coloneqq (0, \infty)$. We denote the inner product in $\mathbb{R}^d$ by $\langle \cdot, \cdot \rangle$ and its induced $\ell_2$-norm by $\| \cdot \|$. For a vector $x \in \mathbb{R}^d$, $[x]_j$ denotes its $j$th coordinate ($j \in [\![d]\!]$). For a function $f \colon \mathbb{R}^d \to \mathbb{R} \cup \{\pm\infty\}$, $\partial_j f$ denotes its partial derivative with respect to the $j$th coordinate, for $j \in [\![d]\!]$. The ceiling function is denoted by $\lceil \cdot \rceil$.

## 3.2 Problem Setting

We consider the problem of minimizing the *expected risk* $\mathbb{E}_{\xi \sim \mathsf{P}}[f(x; \xi)]$ with respect to $x \in \mathbb{R}^d$, where the random variable $\xi \in \mathcal{Z} \subseteq \mathbb{R}^p$ is distributed according to the unknown true data distribution $\mathsf{P}$. We approximate $\mathsf{P}$ by the empirical distribution $\widehat{\mathsf{P}} = \frac{1}{n} \sum_{i=1}^{n} \delta_{\xi_i}$, where $\{\xi_i\}_{i \in [\![n]\!]}$

is a sample of size $n \in \mathbb{N}^*$. This leads to the *empirical risk minimization* problem:

$$\underset{x \in \mathbb{R}^d}{\text{minimize}} \ F(x) := \mathbb{E}_{\xi \sim \widehat{\mathsf{P}}}[f(x;\xi)] = \frac{1}{n} \sum_{i=1}^{n} f(x;\xi_i).$$

When the sample size $n$ is large, the gradient of $F$ is approximated by its batch counterparts. Given a batch of samples $\mathcal{B} \subset [\![n]\!]$ of size $b := |\mathcal{B}|$, we define the batch loss associated with $\mathcal{B}$ by $F_{\mathcal{B}}(x) := \frac{1}{b} \sum_{i \in \mathcal{B}} f(x;\xi_i)$. If $f(\cdot;\xi)$ is continuously differentiable for every $\xi \in \mathcal{Z}$, then the loss and batch-loss gradients are given, respectively, by $\nabla F(x) = \frac{1}{n} \sum_{i=1}^{n} \nabla f(x;\xi_i)$ and $\nabla F_{\mathcal{B}}(x) = \frac{1}{b} \sum_{i \in \mathcal{B}} \nabla f(x;\xi_i)$. The batch gradient is an unbiased estimator of the full gradient, that is, $\mathbb{E}_{\mathcal{B}}[\nabla F_{\mathcal{B}}(x)] = \nabla F(x)$ for every $x \in \mathbb{R}^d$. In many applications, including deep learning, the objective function $F$ is nonconvex. Thus, we consider the problem of finding a (first-order) $\varepsilon$-stationary point $x^\star \in \mathbb{R}^d$ satisfying $\|\nabla F(x^\star)\|^2 \leqslant \varepsilon$, rather than a global minimum, which is generally intractable.

## 4 Adaptive Sampling Methods

The adaptive batch size schemes proposed in this paper are based on a family of *adaptive sampling methods* for stochastic optimization problems. We distinguish throughout between the *exact* population-level conditions used in the convergence analysis and the *approximate* batch-based tests used in implementation. Although such methods have been extended to more general settings (see Section 2 for details), here we focus on the unconstrained case.

### 4.1 Norm Test

Byrd et al. (2012) proposed the *norm test*, also called the *norm condition*, based on the following observation.

**Proposition 1.** *Let $x \in \mathbb{R}^d$ satisfy $\nabla F(x) \neq 0$. If there exists $\eta \in [0,1)$ such that*

$$\delta_{\mathcal{B}}(x) := \|\nabla F_{\mathcal{B}}(x) - \nabla F(x)\| \leqslant \eta \|\nabla F(x)\|, \tag{1}$$

*then $-\nabla F_{\mathcal{B}}(x)$ is a descent direction for $F$ at $x$.*

**Proof** A vector $d$ is a descent direction for $F$ at $x$ if and only if $\langle \nabla F(x), d \rangle < 0$. Thus it suffices to show that $\langle \nabla F_{\mathcal{B}}(x), \nabla F(x) \rangle > 0$. If (1) holds, then

$$\|\nabla F_{\mathcal{B}}(x) - \nabla F(x)\|^2 = \|\nabla F_{\mathcal{B}}(x)\|^2 - 2\langle \nabla F_{\mathcal{B}}(x), \nabla F(x) \rangle + \|\nabla F(x)\|^2 \leqslant \eta^2 \|\nabla F(x)\|^2,$$

which implies

$$2\langle \nabla F_{\mathcal{B}}(x), \nabla F(x) \rangle \geqslant (1-\eta^2)\|\nabla F(x)\|^2 + \|\nabla F_{\mathcal{B}}(x)\|^2 > 0,$$

because $\eta < 1$ and $\nabla F(x) \neq 0$. Therefore $-\nabla F_{\mathcal{B}}(x)$ is a descent direction for $F$ at $x$. ∎

For iteration $k \in \mathbb{N}^*$, we define the filtration $\mathcal{F}_k := \sigma(\{x_1, \mathcal{B}_1, \ldots, \mathcal{B}_{k-1}\})$, which contains the randomness revealed before drawing the batch $\mathcal{B}_k$ at iteration $k$. We abbreviate $\mathbb{E}_k[\cdot] := \mathbb{E}[\cdot \mid \mathcal{F}_k]$.

The exact counterpart used in the convergence analysis is the *exact variance norm test*

$$\mathbb{E}_k\big[\|\nabla F_{\mathcal{B}_k}(x_k) - \nabla F(x_k)\|^2\big] \leqslant \eta^2 \|\nabla F(x_k)\|^2. \tag{2}$$

This is an expectation-based analogue of (1). It is not a pointwise guarantee.

When $\mathcal{B}$ is sampled uniformly without replacement from $[\![n]\!]$, the left-hand side of (2) can be estimated from the batch by

$$\widehat{\delta}_{\mathcal{B}}(x)^2 := \frac{n-b}{nb}\mathrm{Var}_{i \in \mathcal{B}}(\nabla f(x; \xi_i)),$$

where, for any vector-valued function $h\colon \mathbb{R}^d \times \mathcal{Z} \to \mathbb{R}^d$, the sample variance is defined by

$$\mathrm{Var}_{i \in \mathcal{B}}(h(x; \xi_i)) := \frac{1}{b-1}\sum_{i \in \mathcal{B}}\left\| h(x; \xi_i) - \frac{1}{b}\sum_{j \in \mathcal{B}} h(x; \xi_j)\right\|^2. \tag{3}$$

In the large-data regime, or under sampling with replacement, one typically drops the finite-population correction and uses the *approximate norm test*

$$\frac{\mathrm{Var}_{i \in \mathcal{B}_k}(\nabla f(x_k; \xi_i))}{b_k} \leqslant \eta^2 \|\nabla F_{\mathcal{B}_k}(x_k)\|^2. \tag{4}$$

If (4) fails, a common monotone update is

$$b_{k+1} = \max\left\{ b_k, \left\lceil \frac{\mathrm{Var}_{i \in \mathcal{B}_k}(\nabla f(x_k; \xi_i))}{\eta^2 \|\nabla F_{\mathcal{B}_k}(x_k)\|^2}\right\rceil\right\}.$$

This is the smallest new batch size that would satisfy (4) if the estimated variance and batch-gradient norm were unchanged. In practice, one uses the statistics computed from the current batch to choose $b_{k+1}$ and then draws the next batch at that size, without rechecking the test on an enlarged batch at $x_k$.

## 4.2 Inner Product Test

Bollapragada et al. (2018a) observed that the norm test often increases the batch size too aggressively, and proposed the *inner product test* as a milder alternative. In the conditionally i.i.d. sampling model, the *exact variance inner product test* requires that there exists $\vartheta > 0$ such that

$$\frac{1}{b_k}\mathbb{E}_k\Big[\big(\langle \nabla f(x_k; \xi), \nabla F(x_k)\rangle - \|\nabla F(x_k)\|^2\big)^2\Big] \leqslant \vartheta^2 \|\nabla F(x_k)\|^4, \tag{5}$$

where $\xi$ denotes a generic sample drawn according to the batch-sampling scheme at iteration $k$. Under conditional independence, this controls the variance of the projection $\langle \nabla F_{\mathcal{B}_k}(x_k), \nabla F(x_k)\rangle$.

Since $\nabla F(x_k)$ is not available, implementations usually replace it with $\nabla F_{\mathcal{B}_k}(x_k)$ and use the *approximate inner product test*

$$\frac{\mathrm{Var}_{i \in \mathcal{B}_k}(\langle \nabla f(x_k; \xi_i), \nabla F_{\mathcal{B}_k}(x_k)\rangle)}{b_k} \leqslant \vartheta^2 \|\nabla F_{\mathcal{B}_k}(x_k)\|^4, \tag{6}$$

where the variance on the left-hand side is computed using (3).

To rule out the possibility that $\nabla F_{\mathcal{B}_k}(x_k)$ is nearly orthogonal to $\nabla F(x_k)$, Bollapragada et al. (2018a) also introduced the *exact variance orthogonality test*: for $\nabla F(x_k) \neq 0$, there exists $\nu > 0$ such that

$$\frac{1}{b_k}\mathbb{E}_k\left[\left\|\nabla f(x_k; \xi) - \frac{\langle \nabla f(x_k; \xi), \nabla F(x_k)\rangle}{\|\nabla F(x_k)\|^2}\nabla F(x_k)\right\|^2\right] \leqslant \nu^2\|\nabla F(x_k)\|^2. \tag{7}$$

The practical approximation, valid when $\nabla F_{\mathcal{B}_k}(x_k) \neq 0$, is

$$\frac{1}{b_k}\mathrm{Var}_{i\in\mathcal{B}_k}\left(\nabla f(x_k; \xi_i) - \frac{\langle \nabla f(x_k; \xi_i), \nabla F_{\mathcal{B}_k}(x_k)\rangle}{\|\nabla F_{\mathcal{B}_k}(x_k)\|^2}\nabla F_{\mathcal{B}_k}(x_k)\right) \leqslant \nu^2\|\nabla F_{\mathcal{B}_k}(x_k)\|^2, \tag{8}$$

where the variance on the left-hand side is again computed via (3). The conditions (6) and (8) together form the *approximate augmented inner product test*.

If either approximate condition fails, one may update the next batch size by

$$b_{k+1} = \max\Bigg\{b_k, \Bigg\lceil \max\Bigg\{\frac{\mathrm{Var}_{i\in\mathcal{B}_k}(\langle \nabla f(x_k; \xi_i), \nabla F_{\mathcal{B}_k}(x_k)\rangle)}{\vartheta^2\|\nabla F_{\mathcal{B}_k}(x_k)\|^4},$$
$$\frac{\mathrm{Var}_{i\in\mathcal{B}_k}\left(\nabla f(x_k; \xi_i) - \frac{\langle \nabla f(x_k; \xi_i), \nabla F_{\mathcal{B}_k}(x_k)\rangle}{\|\nabla F_{\mathcal{B}_k}(x_k)\|^2}\nabla F_{\mathcal{B}_k}(x_k)\right)}{\nu^2\|\nabla F_{\mathcal{B}_k}(x_k)\|^2}\Bigg\}\Bigg\rceil\Bigg\},$$

and then use a batch of that size at the next iteration. As with the norm test, this update is heuristic and is based on the current batch statistics. The constants $(\vartheta, \nu) \in \mathbb{R}^2_{++}$ must be chosen in practice.

In large-scale implementations, one may also use the inner product test alone, since computing the quantities in (8) introduces additional overhead and near-orthogonality has not been observed in practice in Bollapragada et al. (2018a). In that case, the batch size is updated according to

$$b_{k+1} = \max\Bigg\{b_k, \left\lceil\frac{\mathrm{Var}_{i\in\mathcal{B}_k}(\langle \nabla f(x_k; \xi_i), \nabla F_{\mathcal{B}_k}(x_k)\rangle)}{\vartheta^2\|\nabla F_{\mathcal{B}_k}(x_k)\|^4}\right\rceil\Bigg\}.$$

## 4.3   Adaptive Sampling Methods for Adaptive Gradient Methods

We focus on two simple adaptive gradient methods, ADAGRAD (Duchi et al., 2011; McMahan and Streeter, 2010) and ADAGRAD-NORM (Streeter and McMahan, 2010), whose step sizes are computed adaptively from the magnitudes of previous stochastic gradients.

ADAGRAD was proposed for online convex optimization and takes the form

$$(\forall k \in \mathbb{N}^*) \quad v_k = v_{k-1} + g_k^2, \quad x_{k+1} = x_k - \alpha g_k \odot v_k^{-1/2}, \tag{9}$$

where $g_k \coloneqq \nabla F_{\mathcal{B}_k}(x_k)$, $\alpha > 0$ is a constant step size, $\odot$ denotes the Hadamard product, and the powers are taken coordinate-wise. Since $(v_k)_{k\in\mathbb{N}} \subset \mathbb{R}^d_{++}$, ADAGRAD uses adaptive coordinate-wise step sizes.

ADAGRAD-NORM, also known as SGD with ADAGRAD step sizes, is the scalar variant of

ADAGRAD:

$$(\forall k \in \mathbb{N}^*) \quad v_k = v_{k-1} + \|g_k\|^2, \quad x_{k+1} = x_k - \alpha g_k/\sqrt{v_k}, \tag{10}$$

where $(v_k)_{k\in\mathbb{N}} \subset \mathbb{R}_{++}$ is a sequence of positive scalars. The scalar step size $\alpha/\sqrt{v_k}$ makes ADAGRAD-NORM easier to analyze (Ward et al., 2019, 2020; Faw et al., 2022; Attia and Koren, 2023).

The norm test and the augmented inner product test were originally developed for SGD, but they depend only on the discrepancy between full gradients and batch gradients and are therefore largely optimizer-agnostic. Thus ADADAGRAD and ADADAGRAD-NORM are obtained by combining the same batch-size rules with the updates (9) and (10). The complete pseudocode is given in Algorithm 1.

Whenever a denominator in a test statistic vanishes, the corresponding condition is understood in the limiting sense that it can hold only if the numerator also vanishes.

The coordinate-wise nature of the adaptive step sizes in ADAGRAD motivates a coordinate-wise variant of ADADAGRAD. This leads to the coordinate-wise exact variance norm test: for each $j \in [\![d]\!]$,

$$\mathbb{E}_k\Big[(\partial_j F_{\mathcal{B}_k}(x_k) - \partial_j F(x_k))^2\Big] \leqslant \eta^2 (\partial_j F(x_k))^2, \tag{11}$$

with approximate form

$$\frac{1}{b_k(b_k-1)} \sum_{i\in\mathcal{B}_k} (\partial_j f(x_k; \xi_i) - \partial_j F_{\mathcal{B}_k}(x_k))^2 \leqslant \eta^2 (\partial_j F_{\mathcal{B}_k}(x_k))^2.$$

This coordinate-wise condition is stronger than the scalar norm test and is the one used later to analyze ADADAGRAD. By contrast, the inner product and orthogonality tests intrinsically couple coordinates through inner products, so straightforward coordinate-wise analogues are not immediate.

## 5  Convergence Analysis

We provide two sets of convergence-rate results for ADADAGRAD and ADADAGRAD-NORM. Under $L$-Lipschitz smoothness, we establish convergence-rate results for ADADAGRAD-NORM with either the norm test or the augmented inner product test, and for ADADAGRAD with a coordinate-wise norm test. We then analyze ADADAGRAD-NORM under generalized smoothness. We first record the smoothness assumptions used below.

**Assumption 2** ($L$-Lipschitz smoothness). *The function $F\colon \mathbb{R}^d \to \mathbb{R}$ is continuously differentiable, bounded below by $F^\star := \inf_{x\in\mathbb{R}^d} F(x) \in \mathbb{R}$, and its gradient $\nabla F$ is $L$-Lipschitz continuous with constant $L > 0$, that is, for any $x, y \in \mathbb{R}^d$,*

$$\|\nabla F(x) - \nabla F(y)\| \leqslant L\|x - y\|.$$

The uniform smoothness condition is often excessively restrictive in practice. When $F$ is twice continuously differentiable, Theorem 2 is equivalent to the uniform Hessian bound $\|\nabla^2 F(x)\|_{\mathrm{op}} \leqslant L$ for all $x \in \mathbb{R}^d$ (equivalently, $-LI_d \preccurlyeq \nabla^2 F(x) \preccurlyeq LI_d$). For example, Zhang

**Algorithm 1** Adaptive Batch Size Schemes for (Adaptive) Stochastic Gradient Methods (ADASGD, ADADADAGRAD, and ADADADAGRAD-NORM)

---

**Input:** $x_1 \in \mathbb{R}^d$; $v_0 \in \mathbb{R}_{++}^d$ for ADADADAGRAD or $v_0 \in \mathbb{R}_{++}$ for ADADADAGRAD-NORM; $\mathcal{D} = \{\xi_i\}_{i \in [\![n]\!]} \subset \mathcal{Z}$; either $\eta \in (0, 1)$ (norm test) or $(\vartheta, \nu) \in \mathbb{R}_{++}^2$ (augmented inner product test); total number of processed samples $N \in \mathbb{N}^*$; step counter $k = 1$; processed-samples counter $i = 0$; initial batch size $2 \leqslant b_1 \ll n$; step size sequence $(\alpha_k)_{k \in \mathbb{N}^*}$ for ADASGD or step size $\alpha > 0$ for ADADADAGRAD and ADADADAGRAD-NORM

**while** $i < N$ **do**

    Sample a minibatch $\mathcal{B}_k$ of size $b_k$ from $\mathcal{D}$ according to the chosen sampling scheme

    Compute the batch gradient $g_k := \nabla F_{\mathcal{B}_k}(x_k) = \frac{1}{b_k} \sum_{i \in \mathcal{B}_k} \nabla f(x_k; \xi_i)$

    **if** norm test **then**

        Compute $\mathrm{Var}_{i \in \mathcal{B}_k}(\nabla f(x_k; \xi_i))$

        **if** coordinate-wise **then**

            Compute the coordinate-wise norm test statistics $\mathsf{T}_j(x_k; \mathcal{B}_k, \eta) := \frac{1}{b_k - 1} \sum_{i \in \mathcal{B}_k} (\partial_j f(x_k; \xi_i) - \partial_j F_{\mathcal{B}_k}(x_k))^2 / (\eta^2 (\partial_j F_{\mathcal{B}_k}(x_k))^2), \quad j \in [\![d]\!]$

            Compute the aggregate coordinate-wise norm test statistic $\mathsf{T} = \max_{j \in [\![d]\!]} \mathsf{T}_j$

        **else**

            Compute the norm test statistic $\mathsf{T} \equiv \mathsf{T}(x_k; \mathcal{B}_k, \eta) := \mathrm{Var}_{i \in \mathcal{B}_k}(\nabla f(x_k; \xi_i)) / (\eta^2 \|g_k\|^2)$

        **end if**

    **end if**

    **if** augmented inner product test **then**

        Compute the variance of the inner product between the batch per-sample gradients and the batch gradient $\mathrm{Var}_{i \in \mathcal{B}_k}(\langle \nabla f(x_k; \xi_i), \nabla F_{\mathcal{B}_k}(x_k) \rangle)$

        Compute the inner product test statistic $\mathsf{T}_{\mathrm{ip}}(x_k; \mathcal{B}_k, \vartheta) := \mathrm{Var}_{i \in \mathcal{B}_k}(\langle \nabla f(x_k; \xi_i), \nabla F_{\mathcal{B}_k}(x_k) \rangle) / (\vartheta^2 \|\nabla F_{\mathcal{B}_k}(x_k)\|^4)$

        Compute the variance of the discrepancy of orthogonality between the batch per-sample gradients and the batch gradient

$$\mathsf{V}(x_k; \mathcal{B}_k) := \mathrm{Var}_{i \in \mathcal{B}_k}\left( \nabla f(x_k; \xi_i) - \frac{\langle \nabla f(x_k; \xi_i), \nabla F_{\mathcal{B}_k}(x_k) \rangle}{\|\nabla F_{\mathcal{B}_k}(x_k)\|^2} \nabla F_{\mathcal{B}_k}(x_k) \right)$$

        Compute the orthogonality test statistic

$$\mathsf{T}_{\mathrm{ortho}}(x_k; \mathcal{B}_k, \nu) := \mathsf{V}(x_k; \mathcal{B}_k) / (\nu^2 \|\nabla F_{\mathcal{B}_k}(x_k)\|^2)$$

        Compute the augmented inner product test statistic $\mathsf{T} = \max\{\mathsf{T}_{\mathrm{ip}}, \mathsf{T}_{\mathrm{ortho}}\}$

    **end if**

    $b_{k+1} = \max\{\lceil \mathsf{T} \rceil, b_k\}$

    $x_{k+1} = x_k - \alpha_k \nabla F_{\mathcal{B}_k}(x_k)$                             $\triangleright$ *ADASGD*

    **or**

    $v_k = v_{k-1} + \|\nabla F_{\mathcal{B}_k}(x_k)\|^2$ and $x_{k+1} = x_k - \alpha \nabla F_{\mathcal{B}_k}(x_k) / \sqrt{v_k}$     $\triangleright$ *ADADADAGRAD-NORM*

    **or**

    $v_k = v_{k-1} + \nabla F_{\mathcal{B}_k}(x_k)^2$ and $x_{k+1} = x_k - \alpha \nabla F_{\mathcal{B}_k}(x_k) \odot v_k^{-1/2}$     $\triangleright$ *ADADADAGRAD*

    $i \leftarrow i + b_k$

    $k \leftarrow k + 1$

**end while**

et al. (2020a,b) showed numerically that transformer architectures in language models exhibit loss landscapes that either do not satisfy the Lipschitz smoothness assumption or have very large Lipschitz constants $L$. To address this issue, Zhang et al. (2020b) proposed the following weaker generalized smoothness condition:

**Assumption 3** $((L_0, L_1)$-smoothness**)**. *The function $F\colon \mathbb{R}^d \to \mathbb{R}$ is continuously differentiable, bounded below by $F^\star := \inf_{x \in \mathbb{R}^d} F(x) \in \mathbb{R}$, and satisfies*

$$\|\nabla F(x) - \nabla F(y)\| \leqslant (L_0 + L_1 \|\nabla F(x)\|)\|x - y\|,$$

*for any $x, y \in \mathbb{R}^d$, where $(L_0, L_1) \in \mathbb{R}_+^2$.*

In addition to these smoothness assumptions, the adaptive sampling tests induce a control on the second moment of the stochastic gradient along the iterates. Using the nomenclature of Khaled and Richtárik (2023), we introduce the expected strong growth condition (Vaswani et al., 2019) below.

**Definition 4** (Expected strong growth). *Let $\mathcal{B} \subset [\![n]\!]$ denote a random batch. The* expected strong growth *(E-SG) condition is given by*

$$(\forall x \in \mathbb{R}^d) \qquad \mathbb{E}[\|\nabla F_{\mathcal{B}}(x)\|^2] \leqslant \tau \|\nabla F(x)\|^2$$

*for some constant $\tau > 0$, where the expectation is taken with respect to the batch-sampling randomness. In our analysis, it suffices to require this condition only along the iterates, that is,*

$$(\forall k \in \mathbb{N}^*) \quad \mathbb{E}_k[\|\nabla F_{\mathcal{B}_k}(x_k)\|^2] \leqslant \tau \|\nabla F(x_k)\|^2, \tag{12}$$

*for some constant $\tau > 0$, where $(x_k)_{k \in \mathbb{N}^*}$ are the iterates generated by a stochastic gradient method.*

**Exact-test convention for the convergence analysis.** The convergence results in this section are stated for idealized exact-test versions of the adaptive batch-size schemes. Specifically, at iteration $k$, the batch size is assumed to be chosen so that the corresponding population-level variance condition holds at the current iterate $x_k$ before the stochastic update is taken. This convention is used to isolate the convergence properties of the adaptive-sampling principle from the additional statistical error introduced by estimating the test quantities from a finite minibatch. The empirical algorithms in Section 6 use batch-based approximate tests, where the statistic computed from the current minibatch is used to choose the next batch size. Thus, the results below should be interpreted as convergence guarantees for the exact-test adaptive-sampling principle rather than for every implementation detail of the approximate empirical rule.

**Proposition 5** (Informal). *Suppose that the batch gradient is conditionally unbiased at each iteration. If the exact variance norm test holds with constant $\eta \in (0, 1)$, then the iteration-wise E-SG condition (12) holds with $\tau = 1 + \eta^2$. Likewise, if the samples in $\mathcal{B}_k$ are conditionally*

*i.i.d. and* $\nabla F(x_k) \neq 0$, *then the exact variance augmented inner product test with constants* $(\vartheta, \nu) \in \mathbb{R}^2_{++}$ *implies* (12) *with* $\tau = 1 + \vartheta^2 + \nu^2$.

A more precise statement and its proof are given in Section A; see Theorems 10 and 11. Although the expected strong growth condition is often studied in overparameterized models (Vaswani et al., 2019) and is widely used in deep learning, the constant $\tau$ is usually problem-specific and unknown. In contrast, the adaptive sampling methods only require the corresponding bound to hold along the realized iterates $(x_k)_{k \in \mathbb{N}^*}$, and the resulting value of $\tau$ is explicitly determined by the test parameters. When an approximate test fails, the batch size is increased according to the adaptive-sampling rule from Section 4; under the approximation described there, this is intended to make the test hold for the enlarged batch, rather than to provide a separate guarantee at the next iterate.

**Comparison with existing convergence rates.** Before presenting our main results, we summarize in Table 1 how the proposed guarantees compare with representative convergence results for SGD, adaptive sampling methods, ADAGRAD, and ADAGRAD-NORM on smooth nonconvex objectives. The comparison is qualitative: different results use different stochastic-gradient assumptions, and the rates below should be interpreted together with the corresponding noise or sampling conditions. In particular, the $\mathcal{O}(1/K)$ rates obtained in this paper rely on exact adaptive sampling tests, which imply an expected strong growth condition along the iterates. They should not be interpreted as unconditional improvements over the standard $\mathcal{O}(1/\sqrt{K})$ rates under bounded variance.

Table 1: Comparison of convergence guarantees for stochastic gradient methods on smooth nonconvex objectives. Here, $K$ denotes the number of iterations and $\delta \in (0, 1)$ denotes the failure-probability parameter in the high-probability guarantees; equivalently, the corresponding bounds hold with probability at least $1 - \delta$. The notation $\widetilde{\mathcal{O}}$ hides logarithmic factors and problem-dependent constants.

| Method | Sampling / noise condition | Guarantee type | Rate |
|---|---|---|---|
| SGD | bounded variance | expectation | $\mathcal{O}(K^{-1/2})$ |
| SGD with adaptive sampling | exact adaptive sampling test | expectation | $\mathcal{O}(K^{-1})$ |
| ADAGRAD | standard stochastic gradient assumptions | expectation or high probability | $\widetilde{\mathcal{O}}(K^{-1/2})$ |
| ADAGRAD-NORM | standard stochastic-gradient assumptions | expectation or high probability | $\widetilde{\mathcal{O}}(K^{-1/2})$ |
| ADADAGRAD-NORM | exact norm or augmented inner-product test | high probability | $\mathcal{O}(K^{-1}\delta^{-2})$ |
| ADADAGRAD | coordinate-wise exact norm test | high probability | $\mathcal{O}(K^{-1}\delta^{-2})$ |
| ADADAGRAD-NORM under $(L_0, L_1)$-smoothness | exact norm or augmented inner-product test | high probability | $\mathcal{O}(K^{-1}\delta^{-2})$ |

## 5.1 Convergence Results

We first establish high-probability convergence rates for the adaptive batch size schemes used with ADAGRAD-NORM and ADAGRAD, substantially extending existing convergence-rate results in expectation for SGD; see, e.g., De et al. (2016, 2017); Bollapragada et al. (2018a).

**Technical challenges beyond SGD.** The main difficulty relative to adaptive sampling for SGD is that the stochastic gradient appears both in the update direction and in the adaptive denominator. For ADAGRAD-NORM, the update uses $g_k/\sqrt{v_k}$ with $v_k = v_{k-1} + \|g_k\|^2$, so the random numerator $g_k$ and the random denominator $\sqrt{v_k}$ are coupled. Consequently, conditional unbiasedness of $g_k$ does not directly imply an unbiased descent direction after adaptive normalization. Our analysis handles this coupling by decomposing

$$\left\langle \nabla F(x_k), \mathbb{E}_k\left[\frac{g_k}{\sqrt{v_k}}\right]\right\rangle = \frac{\|\nabla F(x_k)\|^2}{\sqrt{v_{k-1}}} + \left\langle \nabla F(x_k), \mathbb{E}_k\left[\left(\frac{1}{\sqrt{v_k}} - \frac{1}{\sqrt{v_{k-1}}}\right)g_k\right]\right\rangle,$$

and bounding the second term through a potential-difference argument. The remaining second-order terms are controlled by logarithmic ADAGRAD-type sums such as $\sum_{k=1}^{K}\|g_k\|^2/v_k \leqslant \log(v_K) - \log(v_0)$. For coordinate-wise ADAGRAD, the same issue occurs separately in each coordinate, which motivates the coordinate-wise norm test used in the convergence theorem.

ADAGRAD-NORM combined with either the norm test or the augmented inner product test, with any constant initial step size $\alpha > 0$, enjoys the following high-probability convergence bound for nonconvex functions. Before stating the convergence result for ADADAGRAD-NORM, we emphasize that the theorem applies to the exact variance tests : the exact variance norm test or the augmented inner product test is assumed to hold at the iterate $x_k$ used for the update. The batch-based implementation used in the experiments is an approximate version of this idealized rule.

**Theorem 6** (ADADAGRAD-NORM)**.** *Suppose that Theorem 2 holds. Let $(x_k)_{k\in\mathbb{N}^*}$ be the ADAGRAD-NORM iterates (10) with any step size $\alpha > 0$, where the batch sizes $(b_k)_{k\in\mathbb{N}^*}$ are chosen so that either the (exact variance) norm test (2) with constant $\eta \in (0,1)$ or the (exact variance) augmented inner product test (5) and (7) with constants $(\vartheta, \nu) \in \mathbb{R}_{++}^2$ is satisfied at each iteration. Fix $K \in \mathbb{N}^*$, $\delta \in (0,1)$, and $\rho \in (1,\infty)$, and set*

$$\tau := \begin{cases} 1 + \eta^2, & \text{for the norm test,} \\ 1 + \vartheta^2 + \nu^2, & \text{for the augmented inner product test.} \end{cases}$$

*Define*

$$c_1 := \frac{2}{\alpha(1-\rho^{-1})}\left(F(x_1) - F^\star + \frac{\tau\alpha\|\nabla F(x_1)\|^2}{2\sqrt{v_0}} + \frac{\tau L^2\alpha^3(1+\rho\tau)}{\sqrt{v_0}} - \frac{L\alpha^2}{2}\log(v_0)\right),$$

$$c_1^+ := \max\{c_1, 0\}, \qquad c_2 := \frac{L\alpha}{1-\rho^{-1}}, \qquad c_3 := 2\sqrt{v_0} + 2\tau c_1^+ + 8\tau c_2\log(\tau c_2 + 1).$$

*Then, with probability at least $1 - \delta$, we have*

$$\min_{k\in[\![K]\!]}\|\nabla F(x_k)\|^2 \leqslant \frac{c_3(c_1^+ + 2c_2\log c_3)}{K\delta^2}.$$

The proof of Theorem 6 is technical, and its full details are given in Section A.2.1.

**Choice of $\alpha$ and $\rho$ in Theorem 6.** Theorem 6 holds for any constant $\alpha > 0$ under $L$-Lipschitz

smoothness; no upper bound on $\alpha$ is required for the validity of the statement. However, the constants in the bound depend on $\alpha$, and an excessively large $\alpha$ may make the bound loose or practically uninformative. The parameter $\rho > 1$ is not an algorithmic parameter. It is introduced only in the proof through Young-type inequalities and controls a trade-off among the constants. For a simple interpretation one may fix, for example, $\rho = 2$; optimizing $\rho$ can slightly improve the displayed constant but does not affect the asymptotic rate.

To prove the convergence of ADAGRAD, which has coordinate-wise adaptive step sizes, we need the E-SG condition to hold coordinate-wise along the iterates as well. This indeed holds when we invoke a coordinate-wise version of the norm test, as stated in the following proposition.

**Proposition 7** (Coordinate-wise expected strong growth). *Suppose that, for every iteration $k \in \mathbb{N}^*$, the batch gradient $\nabla F_{\mathcal{B}_k}(x_k)$ is conditionally unbiased, that is,*

$$\mathbb{E}_k[\nabla F_{\mathcal{B}_k}(x_k)] = \nabla F(x_k),$$

*and the coordinate-wise exact variance norm test* (11) *holds with constant $\eta \in (0, 1)$. Then the coordinate-wise E-SG condition holds at each iteration, that is, for all $(k, j) \in \mathbb{N}^* \times [\![d]\!]$,*

$$\mathbb{E}_k\left[(\partial_j F_{\mathcal{B}_k}(x_k))^2\right] \leqslant (1 + \eta^2)(\partial_j F(x_k))^2. \tag{13}$$

**Proof** Fix $(k, j) \in \mathbb{N}^* \times [\![d]\!]$. Since

$$\mathbb{E}_k[\partial_j F_{\mathcal{B}_k}(x_k)] = \partial_j F(x_k),$$

we have

$$\mathbb{E}_k\left[(\partial_j F_{\mathcal{B}_k}(x_k) - \partial_j F(x_k))^2\right] = \mathbb{E}_k\left[(\partial_j F_{\mathcal{B}_k}(x_k))^2\right] - (\partial_j F(x_k))^2.$$

Combining this identity with (11) yields (13). ∎

Then ADAGRAD with the coordinate-wise norm test enjoys a similar high-probability convergence guarantee. For ADADAGRAD, the following result is proved under the coordinate-wise exact variance norm test. This condition is stronger than the standard norm test used in our neural network experiments, but it is aligned with the coordinate-wise adaptive preconditioner in ADAGRAD and yields the coordinate-wise expected strong-growth control needed in the proof.

**Theorem 8** (ADADAGRAD). *Suppose that Theorem 2 holds. Let $(x_k)_{k \in \mathbb{N}^*}$ be the ADAGRAD iterates* (9) *with step size $\alpha > 0$, where the batch gradients $g_k := \nabla F_{\mathcal{B}_k}(x_k)$ are conditionally unbiased,*

$$\mathbb{E}_k[g_k] = \nabla F(x_k), \qquad k \in \mathbb{N}^*,$$

*and the batch sizes $(b_k)_{k \in \mathbb{N}^*}$ are chosen so that the coordinate-wise exact variance norm test* (11) *holds with some $\eta \in (0, 1)$ at each iteration. Then there exists a constant $C > 0$, independent of $K \in \mathbb{N}^*$ and $\delta \in (0, 1)$, such that*

$$\mathbb{P}\left(\min_{k \in [\![K]\!]} \|\nabla F(x_k)\|^2 \leqslant \frac{C}{K\delta^2}\right) \geqslant 1 - \delta.$$

The above theorem establishes a sublinear convergence rate (with high probability) for ADADAGRAD on nonconvex functions, whereas such a rate in expectation was established for SGD in Bollapragada et al. (2018a), Theorem 3.4.

**Scope of the ADADAGRAD guarantee.** Theorem 8 applies to ADAGRAD combined with the coordinate-wise exact variance norm test. This condition is natural for the coordinate-wise preconditioner in ADAGRAD and yields a coordinate-wise expected strong-growth bound. In numerical experiments (Section 6), however, evaluating the coordinate-wise test is computationally expensive, and we therefore use the standard norm test as a practical approximation. Consequently, the ADADAGRAD experiments should be viewed as empirical evidence for the practical standard-norm-test variant, rather than as a direct numerical verification of Theorem 8.

Finally, after relaxing the uniform smoothness assumption, ADADAGRAD-NORM still converges under $(L_0, L_1)$-smoothness. When $L_1 = 0$, this reduces to the Lipschitz-smooth case covered by Theorem 6; below we therefore assume $L_1 > 0$. In this regime, the scale parameter $\alpha$ must be upper bounded, and hence one needs knowledge of $L_1$. The same exact-test convention is used in the generalized-smoothness setting below.

**Theorem 9** (($L_0, L_1$)-smooth ADADAGRAD-NORM). *Suppose that Theorem 3 holds with $L_1 > 0$. Let $(x_k, v_k)_{k \in \mathbb{N}^*}$ be generated by (10) from some $v_0 \in \mathbb{R}_{++}$, fix $K \in \mathbb{N}^*$ and $\delta \in (0, 1)$, and define*

$$g_k := \nabla F_{\mathcal{B}_k}(x_k), \qquad k \in [\![K]\!].$$

*(i) for every $k \in [\![K]\!]$, $g_k$ is conditionally unbiased and the exact variance norm test (2) holds with some $\eta \in (0, 1)$, in which case $\tau := 1 + \eta^2$; or*

*(ii) for every $k \in [\![K]\!]$, the samples in $\mathcal{B}_k$ are conditionally i.i.d., $\nabla F(x_k) \neq 0$, and the exact variance inner product test (5) and exact variance orthogonality test (7) hold with some $(\vartheta, \nu) \in \mathbb{R}_{++}^2$, in which case $\tau := 1 + \vartheta^2 + \nu^2$.*

*Let $(\rho_1, \rho_2, \omega) \in \mathbb{R}_{++}^3$ satisfy*

$$\frac{1}{\rho_1} + \frac{\rho_1}{\rho_2} + 2\omega < 1,$$

*and suppose that*

$$\alpha \leqslant \frac{1}{L_1} \min\left\{ \frac{\omega}{2\rho_1 \tau}, \sqrt{\frac{\omega}{2\rho_1 \tau}} \right\}.$$

*Then there exists a constant $C > 0$, depending only on the problem and algorithm parameters but independent of $K$ and $\delta$, such that*

$$\mathbb{P}\left( \min_{k \in [\![K]\!]} \|\nabla F(x_k)\|^2 \leqslant \frac{C}{K\delta^2} \right) \geqslant 1 - \delta.$$

If $\nabla F(x_k) = 0$ for some $k \in [\![K]\!]$, then the conclusion is immediate. The same proof strategy suggests a corresponding generalized-smoothness result for ADAGRAD, but we do not state it here.

## 5.2 Outlines of Proofs

We now provide outlines of the proofs of the above theorems. Full proofs can be found in Section A.

**Proof** [Theorem 6] The proof first invokes the iteration-wise E-SG bound furnished by the exact test:

$$\mathbb{E}_k[\|g_k\|^2] \leqslant \tau\|\nabla F(x_k)\|^2,$$

where $\tau = 1 + \eta^2$ for the norm test and $\tau = 1 + \vartheta^2 + \nu^2$ for the augmented inner product test. Using the $L$-Lipschitz smoothness of $F$, we obtain

$$\mathbb{E}_k[F(x_{k+1})] \leqslant F(x_k) - \alpha\left\langle \nabla F(x_k), \mathbb{E}_k\left[\frac{g_k}{\sqrt{v_k}}\right]\right\rangle + \frac{L\alpha^2}{2}\mathbb{E}_k\left[\left\|\frac{g_k}{\sqrt{v_k}}\right\|^2\right].$$

The inner product is decomposed as

$$\left\langle \nabla F(x_k), \mathbb{E}_k\left[\frac{g_k}{\sqrt{v_k}}\right]\right\rangle = \frac{\|\nabla F(x_k)\|^2}{\sqrt{v_{k-1}}} + \left\langle \nabla F(x_k), \mathbb{E}_k\left[\left(\frac{1}{\sqrt{v_k}} - \frac{1}{\sqrt{v_{k-1}}}\right)g_k\right]\right\rangle.$$

The second term is then controlled by a one-step potential difference $\varphi_{k-1} - \varphi_k$, where $\varphi_k = \|\nabla F(x_k)\|^2/\sqrt{v_k}$, together with a summable remainder of order $\|g_{k-1}\|^2/v_{k-1}^{3/2}$. After summing over $k$, the second-order term is handled through the logarithmic bound $\sum_{k=1}^{K}\|g_k\|^2/v_k \leqslant \log v_K - \log v_0$, which yields, via Jensen's inequality, a uniform bound on $\mathbb{E}[\sqrt{v_K}]$. This in turn bounds $\sum_{k=1}^{K}\mathbb{E}[\|\nabla F(x_k)\|^2/\sqrt{v_{k-1}}]$, and a final Cauchy–Schwarz and Markov argument gives the stated high-probability estimate for $\min_{k\in[\![K]\!]}\|\nabla F(x_k)\|^2$. ∎

**Proof** [Theorem 8] The proof follows the same scheme, but coordinate-wise. Using the coordinate-wise exact variance norm test together with the conditional unbiasedness of $g_k$, one obtains

$$\mathbb{E}_k[g_{k,j}^2] \leqslant (1 + \eta^2)(\partial_j F(x_k))^2, \qquad j \in [\![d]\!].$$

The descent inequality becomes

$$\mathbb{E}_k[F(x_{k+1})] \leqslant F(x_k) - \alpha\sum_{j=1}^{d}\frac{(\partial_j F(x_k))^2}{\sqrt{v_{k-1,j}}} + \alpha\sum_{j=1}^{d}T_{k,j} + \frac{L\alpha^2}{2}\mathbb{E}_k\left[\left\|\frac{1}{\sqrt{v_k}}\odot g_k\right\|^2\right],$$

where

$$T_{k,j} := \partial_j F(x_k)\mathbb{E}_k\left[\left(\frac{1}{\sqrt{v_{k-1,j}}} - \frac{1}{\sqrt{v_{k,j}}}\right)g_{k,j}\right].$$

Each $T_{k,j}$ is bounded by a coordinate-wise potential difference plus a summable remainder. Summing over $j$ and then over $k$ yields control of

$$\sum_{k=1}^{K}\sum_{j=1}^{d}\mathbb{E}\left[\frac{(\partial_j F(x_k))^2}{\sqrt{v_{k-1,j}}}\right].$$

The accumulated second-order terms are handled through the logarithmic bounds $\sum_{k=1}^{K} g_{k,j}^2/v_{k,j} \leqslant \log v_{K,j} - \log v_{0,j}$, which then control $\sum_{j=1}^{d} \mathbb{E}[\log v_{K,j}]$ and hence $\mathbb{E}[\sum_{j=1}^{d} \sqrt{v_{K,j}}]$. The high-probability bound follows from the same final Cauchy–Schwarz and Markov argument as in the proof of Theorem 6. ∎

**Proof** [Theorem 9] The proof parallels that of Theorem 6, but begins with the generalized descent lemma. The step-size restriction ensures $L_1\|x_{k+1} - x_k\| \leqslant 1$, so

$$\mathbb{E}_k[F(x_{k+1})] \leqslant F(x_k) - \alpha\left\langle \nabla F(x_k), \mathbb{E}_k\left[\frac{g_k}{\sqrt{v_k}}\right]\right\rangle + \frac{\alpha^2}{2}(L_0 + L_1\|\nabla F(x_k)\|)\mathbb{E}_k\left[\left\|\frac{g_k}{\sqrt{v_k}}\right\|^2\right].$$

As before, we split the inner product into a main descent term and an error term. The error term and the additional $L_1$-dependent second-order contribution are then combined and bounded by using the E-SG estimate, the generalized smoothness inequality, and the parameter restrictions involving $(\rho_1, \rho_2, \omega)$. This yields a descent inequality of the same general form as in the Lipschitz-smooth case, up to modified constants. Summing over $k$, controlling $\mathbb{E}[\sqrt{v_K}]$ through $\sum_{k=1}^{K} \|g_k\|^2/v_k \leqslant \log v_K - \log v_0$ and Jensen's inequality, and finally applying Cauchy–Schwarz and Markov's inequality, we obtain the claimed bound $\min_{k\in\llbracket K\rrbracket} \|\nabla F(x_k)\|^2 = \mathcal{O}(1/(K\delta^2))$ with high probability. ∎

# 6    Numerical Experiments

We evaluate the performance of the norm test and the (augmented) inner product test with ADAGRAD(-NORM) and SGD for image classification, employing logistic regression (Section 6.1) and a three-layer CNN on the MNIST dataset (LeCun et al., 1998), as well as a three-layer CNN and RESNET-18 (He et al., 2016) on the CIFAR-10 dataset (Krizhevsky, 2009). We note that training larger models often requires multiple workers and data parallelism, such as Distributed Data Parallel (DDP; Li et al., 2020) and Fully Sharded Data Parallel (FSDP; Zhao et al., 2023). Extending and implementing our proposed schemes under data parallelism presents additional complexities and remains an area for future research (see e.g., Lau et al., 2025, for recent results). We thus concentrate on smaller models and datasets with the goal of demonstrating the concept rather than achieving state-of-the-art results. Given the typically large number of parameters $d$, conducting coordinate-wise norm tests for ADADAGRAD is not computationally practical, so the standard norm test is applied. For the purpose of numerical comparison, we also empirically evaluate the norm test with ADAM (Kingma and Ba, 2015) for training RESNET-18 on the CIFAR-10 dataset, which is coined ADADAM.

For the implementation of the proposed schemes, per-sample gradients are computed using JAX-like composable function transforms (Bradbury et al., 2018) called `torch.func` in PyTorch 2.0+. Numerical experiments are carried out on workstations with NVIDIA RTX 2080Ti 11GB (for MNIST), A100 80GB (for CNN on CIFAR-10) and L40S 48GB (for RESNET-18 on CIFAR-10) GPUs, with PyTorch 2.2.1 (Paszke et al., 2019) and Lightning Fabric 2.2.0 (Falcon and The

PyTorch Lightning team, 2019). The ADAGRAD-NORM implementation is taken from Ward et al. (2019). The code for the full implementation of the proposed schemes and reproducing the following training results is available at `https://github.com/timlautk/AdAdaGrad`. Further details of the experiments, such as the hyperparameter setting, and additional results can be found in Section B.

## 6.1 Multi-class Logistic Regression on MNIST

We first apply our methods to a ten-class logistic-regression problem on the MNIST dataset, which has a smooth convex objective. Our experiments are conducted with an equal training budget of 6 million samples (equivalent to 100 epochs), setting a maximum batch size of 60,000 (i.e., the full batch) for all approaches. To highlight the adaptivity and flexibility of our proposed methods, we refrain from conducting an exhaustive search for optimal values of $\alpha$, $\eta$, and $\vartheta$, and we do not use learning-rate schedules across methods. The outcomes, including the number of iterations required (steps), average batch sizes (batch size), final training loss (loss), and final validation accuracy (accuracy), are reported in Table 2.

Table 2: Multi-class logistic regression on MNIST

| Scheme | test | steps | batch size | loss | accuracy |
|---|---|---|---|---|---|
| ADASGD | $\eta = 0.10$ | 351 | 17131 | 1.04 | 0.82 |
| ADASGD | $\eta = 0.25$ | 1029 | 5831 | 0.67 | 0.86 |
| ADADAGRAD-NORM | $\eta = 0.10$ | 596 | 10060 | 1.50 | 0.78 |
| ADADAGRAD-NORM | $\eta = 0.25$ | 2462 | 2437 | 1.02 | 0.83 |
| ADADAGRAD | $\eta = 0.10$ | **126** | **47282** | **0.54** | **0.88** |
| ADADAGRAD | $\eta = 0.25$ | **274** | **21918** | **0.46** | **0.90** |
| ADASGD | $\vartheta = 0.01$ | 717 | 8362 | 0.76 | 0.85 |
| ADASGD | $\vartheta = 0.05$ | 1804 | 3327 | 0.54 | 0.88 |
| ADADAGRAD-NORM | $\vartheta = 0.01$ | 1127 | 5322 | 1.28 | 0.80 |
| ADADAGRAD-NORM | $\vartheta = 0.05$ | 5349 | 1122 | 0.80 | 0.85 |

The results, together with the graphical representations in Figure 1, show that ADADAGRAD, using the norm test with $\eta = 0.25$, outperforms the other methods in terms of both training loss and validation accuracy. These findings underscore the importance of adaptive, and coordinate-wise, learning rates within adaptive sampling methods. Despite the best overall performance of ADADAGRAD with the norm test and $\eta = 0.25$, the variant with $\eta = 0.10$ is more hardware-efficient, with an average batch size exceeding 47,000 and requiring only 126 steps (i.e., gradient evaluations) to process all 6 million samples. This indicates an intrinsic trade-off between computational efficiency and model performance as measured by generalization. Interestingly, ADADAGRAD-NORM, although theoretically simpler to analyze, may underperform relative to ADASGD (i.e., SGD with adaptive batch size schemes) for this specific convex problem and hyperparameter setting. It is also worth noting that the norm test tends to increase batch sizes more aggressively than the inner product test, leading to more efficient use of available GPU memory. We also observe from Figure 1 that, under the norm test, the batch sizes of ADADAGRAD grow much faster than those of ADASGD. We leave a systematic study and theoretical explanation of this phenomenon for future work.

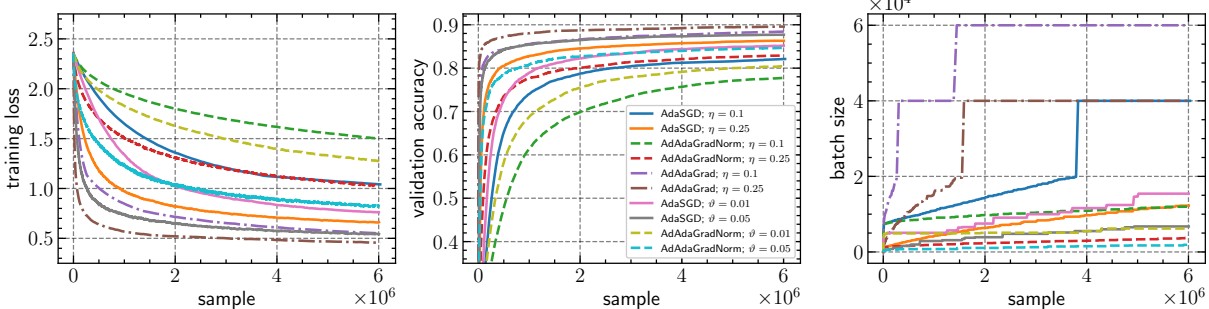

Figure 1: Training loss, validation accuracy, and batch-size curves (vs. number of training samples) of ADASGD, ADADAGRAD, and ADADAGRAD-NORM for logistic regression on the MNIST dataset.

## 6.2 Three-layer CNN on MNIST

We next turn to the application of the proposed methods to training deep neural networks. Specifically, we train a three-layer CNN on the MNIST classification problem, which has a nonconvex objective. Our experiments are conducted with an equal training budget of 6 million samples (equivalent to 100 epochs), setting a maximum batch size of 60,000 (i.e., the full batch) for all approaches. We measure training efficiency by the number of gradient steps rather than wall-clock time, which is device dependent (Shallue et al., 2019). Adaptive batch size methods begin with a small batch size of 8 and gradually increase to the maximum possible batch size of 60,000 (full batch). In Figure 2, ADADAGRAD outperforms ADADAGRAD-NORM and ADASGD in validation accuracy (generalization) by a clear margin. ADADAGRAD-NORM achieves performance similar to that of ADASGD, despite having slightly higher training loss. We also observe from Table 3 that ADADAGRAD using the norm test with $\eta = 0.1$ achieves a validation accuracy of 96% with only 149 iterations and an average batch size of more than 40,000. It uses full batches for the last 70% of its training budget, taking full advantage of the available GPU memory. Referring to Table 3, we observe the generalization gap between small constant batch sizes and large batch sizes, whereas using small batch sizes requires substantially more training time and a larger number of steps, leading to lower training efficiency. Our proposed methods are capable of balancing this trade-off between training efficiency and generalization by introducing adaptive batch size schemes, without the need for extensive tuning of learning rates or pre-specified learning-rate schedules.

We also compare the use of the norm test for SGD and ADAGRAD. For this nonconvex problem, adaptive learning rates become crucial, as ADAGRAD converges much faster than SGD with constant batch sizes. Using the norm test, we observe from Figure 2 that ADADAGRAD increases batch sizes more aggressively than ADASGD, indicating that shorter training time is required (see also Table 3). This suggests the usefulness of adaptive batch size schemes based on the norm test for training nonconvex deep neural networks.

**Choice of constant-batch-size baselines.** We include constant-batch-size baselines to compare the adaptive-batch methods against nonadaptive choices in comparable compute regimes. Since the proposed methods are designed to increase the batch size when the empirical variance

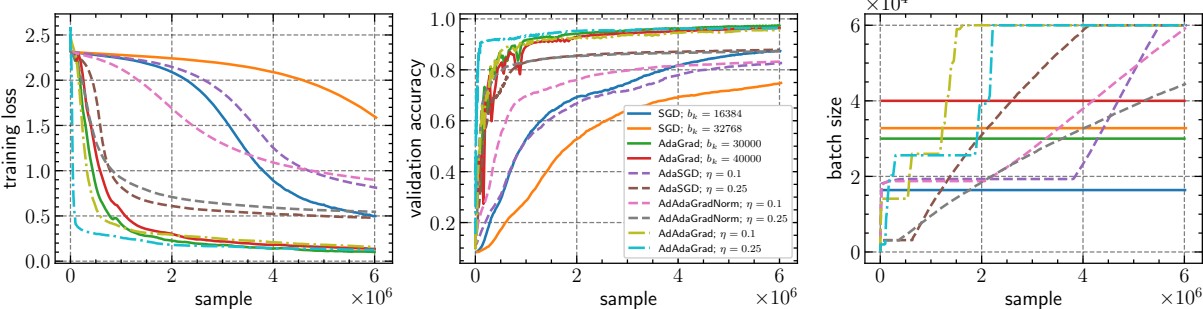

Figure 2: Training loss, validation accuracy, and batch sizes of AdaSGD, AdAdaGrad, and AdAdaGrad-Norm for a three-layer CNN on the MNIST dataset.

test indicates that larger batches are beneficial, the most informative nonadaptive baselines are constant batch sizes in the large-batch regime reached by the adaptive methods. We therefore focus on constant batch sizes that span the practical range of batch sizes observed in the adaptive runs, rather than very small historical defaults such as 128 or 256. This comparison tests whether the adaptive rule can automatically select an effective large-batch regime without fixing the batch size in advance.

Table 3: Three-layer CNN on MNIST

| Scheme | test | steps | time (h) | batch size | loss | accuracy |
|---|---|---|---|---|---|---|
| SGD | N/A | 2929 | 6.77 | 2048 | 0.12 | 0.96 |
| SGD | N/A | 1464 | 3.38 | 4096 | 0.20 | 0.94 |
| SGD | N/A | 732 | 1.72 | 8192 | 0.32 | 0.91 |
| SGD | N/A | 366 | 0.90 | 16384 | 0.51 | 0.87 |
| SGD | N/A | 183 | 0.45 | 32768 | 1.54 | 0.75 |
| SGD | N/A | 99 | 0.27 | 60000 | 2.15 | 0.66 |
| AdaGrad | N/A | 2929 | 7.12 | 2048 | 0.02 | 0.99 |
| AdaGrad | N/A | 1464 | 3.60 | 4096 | 0.02 | 0.99 |
| AdaGrad | N/A | 732 | 1.82 | 8192 | 0.05 | 0.98 |
| AdaGrad | N/A | 366 | 0.92 | 16384 | 0.07 | 0.98 |
| AdaGrad | N/A | 199 | 0.52 | 30000 | 0.10 | 0.97 |
| AdaGrad | N/A | 183 | 0.47 | 32768 | 0.11 | 0.97 |
| AdaGrad | N/A | 149 | 0.36 | 40000 | 0.13 | 0.96 |
| AdaGrad | N/A | 99 | 0.29 | 60000 | 0.17 | 0.95 |
| AdaSGD | $\eta = 0.10$ | 256 | 0.73 | 23546 | 0.79 | 0.83 |
| AdaSGD | $\eta = 0.25$ | 383 | 1.05 | 15627 | 0.48 | 0.88 |
| AdAdaGrad-Norm | $\eta = 0.10$ | 226 | 0.65 | 26567 | 0.88 | 0.83 |
| AdAdaGrad-Norm | $\eta = 0.25$ | 435 | 1.27 | 13830 | 0.54 | 0.87 |
| AdAdaGrad | $\eta = 0.10$ | **149** | **0.45** | **40057** | **0.15** | **0.96** |
| AdAdaGrad | $\eta = 0.25$ | **198** | **0.58** | **30152** | **0.13** | **0.97** |
| AdAdaGrad | $\eta = 0.50$ | **215** | **0.62** | **27940** | **0.11** | **0.97** |
| AdAdaGrad | $\eta = 0.75$ | **271** | **0.79** | **22228** | **0.10** | **0.97** |
| AdaSGD | $\vartheta = 0.01$ | 230 | 0.63 | 26078 | 0.98 | 0.80 |
| AdaSGD | $\vartheta = 0.05$ | 411 | 1.17 | 14593 | 0.45 | 0.88 |
| AdAdaGrad-Norm | $\vartheta = 0.01$ | 241 | 0.70 | 24872 | 0.83 | 0.84 |
| AdAdaGrad-Norm | $\vartheta = 0.05$ | 528 | 1.44 | 11365 | 0.50 | 0.88 |

## 6.3 Three-layer Convolutional Neural Network on CIFAR-10

We next consider a similar problem involving a three-layer CNN with a slightly different architecture on the more challenging CIFAR-10 dataset. We use a training budget of 5 million samples (100 epochs) and a maximum batch size of 10,000 samples. To reduce the computational overhead introduced by the tests, we perform the test every 10 steps. In Figure 3, we again observe that AdaSGD converges more slowly than AdAdaGrad and AdAdaGrad-Norm. This may be due to the lack of a well-crafted learning-rate scaling rule with respect to batch size (cf. the *scaling rule*) for SGD in the nonconvex case: the rapid increase in batch size implies a very small effective learning rate, equal to the ratio of the learning rate to the batch size. Without proper rescaling of the learning rate, such a small effective learning rate could slow convergence. We can empirically support this interpretation because AdaSGD using the inner product test with $\vartheta = 0.1$ increases its batch size slowly and eventually plateaus below 4,000. It also converges much faster than its SGD counterparts, with a final validation accuracy of 57%, approaching the performance of the adaptive methods. We point out, however, that this instance of AdaSGD has a much smaller average batch size and therefore requires many more gradient updates than the adaptive methods under an equal budget of training samples.

Table 4: Three-layer CNN on CIFAR-10

| Scheme | test | steps | batch size | loss | accuracy |
|---|---|---|---|---|---|
| AdaSGD | $\eta = 0.25$ | 523 | 9544 | 1.68 | 0.40 |
| AdaSGD | $\eta = 0.50$ | 658 | 7592 | 1.59 | 0.43 |
| AdAdaGrad-Norm | $\eta = 0.25$ | **531** | **9401** | **1.36** | **0.52** |
| AdAdaGrad-Norm | $\eta = 0.50$ | **1261** | **3964** | **1.19** | **0.57** |
| AdAdaGrad | $\eta = 0.25$ | 903 | 5533 | 1.20 | 0.54 |
| AdAdaGrad | $\eta = 0.50$ | **1123** | **4451** | **1.11** | **0.57** |
| AdaSGD | $\vartheta = 0.05$ | **1597** | **3130** | **1.17** | **0.57** |
| AdaSGD | $\vartheta = 0.10$ | 640 | 7806 | 1.64 | 0.41 |
| AdAdaGrad-Norm | $\vartheta = 0.05$ | **780** | **6413** | **1.29** | **0.55** |
| AdAdaGrad-Norm | $\vartheta = 0.10$ | 1948 | 2567 | 1.13 | 0.58 |

## 6.4 ResNet-18 on CIFAR-10

We finally train a larger network, ResNet-18, for image classification on the CIFAR-10 dataset. We use a training budget of 10 million samples (200 epochs) and a maximum batch size of 50,000 samples. Although the focus of this work is on AdaGrad, we also empirically study the effect of adaptive batch size schemes for Adam because of its ubiquity in deep learning; we refer to this variant as AdAdam. The convergence guarantees for AdAdam are studied in Lau et al. (2025).

**AdAdaGrad.** In Figure 4, we observe that we need to choose a rather small $\eta$ in order to use full batches during later stages of training for this larger model. Comparing AdaGrad with a constant batch size of 50,000 and AdAdaGrad with $\eta = 0.025$ (see also Table 5), AdAdaGrad is able to use full batches in most of the later stages of training while achieving high accuracy with only 23 additional steps. More generally, our proposed scheme again narrows the generalization

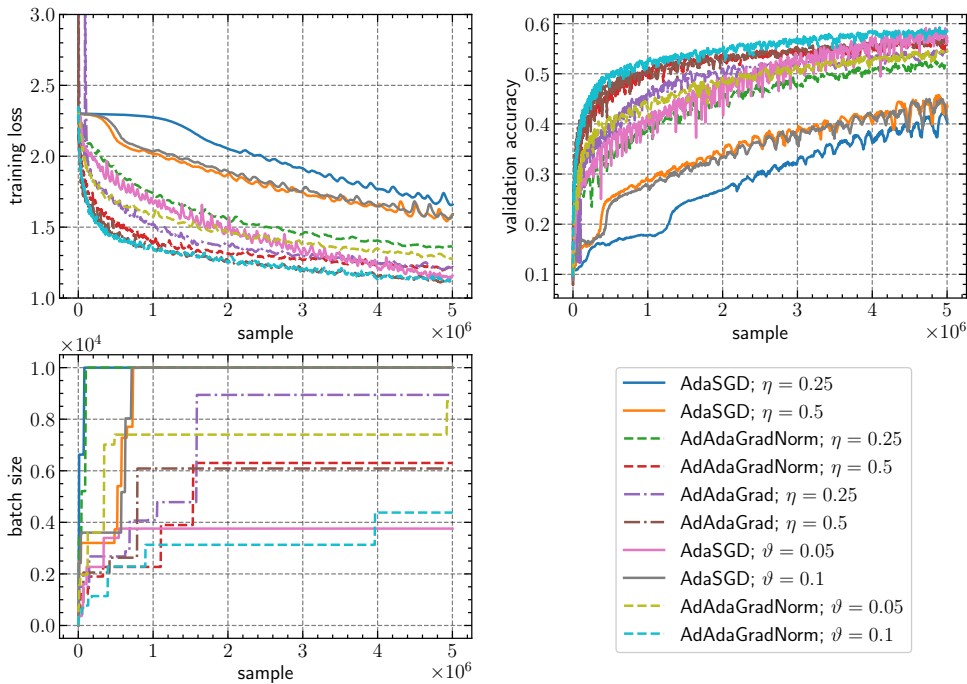

Figure 3: Training loss, validation accuracy, and batch-size curves (vs. number of training samples) of ADASGD, ADADAGRAD, and ADADAGRAD-NORM for a three-layer CNN on the CIFAR-10 dataset.

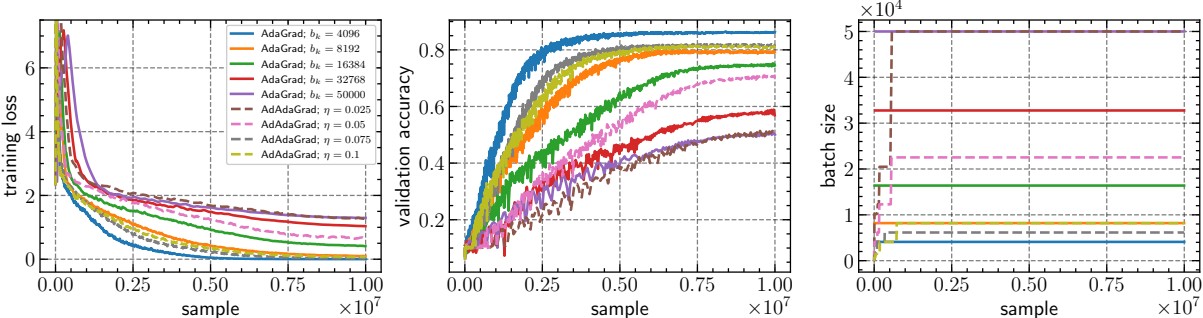

Figure 4: ADAGRAD and ADADAGRAD for RESNET-18 on the CIFAR-10 dataset.

gap between smaller and larger constant batch sizes (e.g., the curves for $\eta = 0.075$ and $0.1$ lie in the gap between those for constant batch sizes 4096 and 8192).

**ADADAM.** In Figure 5, we observe a trend for ADADAM similar to that of ADADAGRAD, but with faster convergence and larger batch-size increases. It is worth noting from Table 5 that ADADAM with $\eta = 0.1$ and an average batch size of 8880 outperforms ADAM with the smaller constant batch size of 8192 in validation accuracy, while requiring almost 100 fewer steps. This suggests that our proposed scheme may be even more beneficial when combined with ADAM.

## 6.5 Discussion

From the numerical experiments, we can draw several interesting conclusions. Adaptive batch size schemes are generally optimizer-agnostic, indicating their broad applicability. In particular, coupling adaptive batch sizes with adaptive gradient optimizers can deliver the best of both worlds: we can narrow the generalization gap while still benefiting from the faster convergence of adaptive

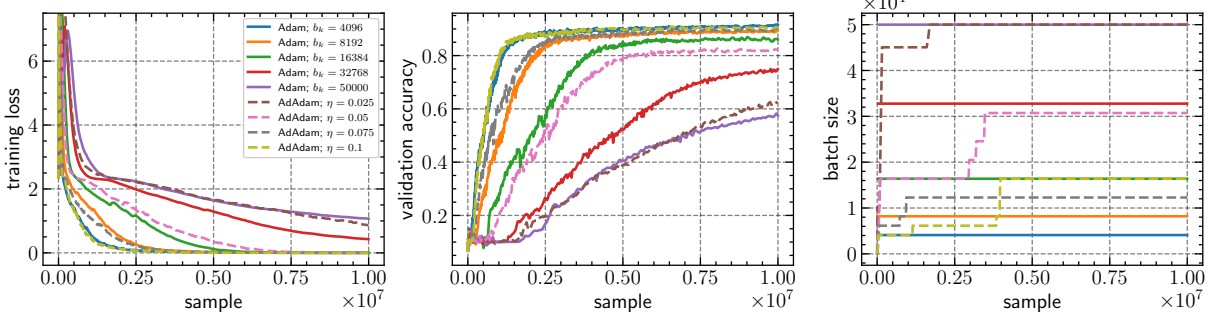

Figure 5: ADAM and ADADAM for RESNET-18 on the CIFAR-10 dataset.

Table 5: RESNET-18 on CIFAR-10

| Scheme | test | steps | time (h) | batch size | loss | accuracy |
|---|---|---|---|---|---|---|
| ADAGRAD | N/A | 2441 | 0.88 | 4096 | 0.0042 | 0.8521 |
| ADAGRAD | N/A | 1220 | 0.70 | 8192 | 0.0808 | 0.8072 |
| ADAGRAD | N/A | 610 | 0.56 | 16384 | 0.5098 | 0.7264 |
| ADAGRAD | N/A | 305 | 0.32 | 32768 | 0.9684 | 0.5816 |
| ADAGRAD | N/A | 199 | 0.23 | 50000 | 1.3625 | 0.4708 |
| ADAM | N/A | 2441 | 1.20 | 4096 | 0.0003 | 0.9147 |
| ADAM | N/A | 1220 | 0.97 | 8192 | 0.0004 | 0.8946 |
| ADAM | N/A | 610 | 0.77 | 16384 | 0.0028 | 0.8628 |
| ADAM | N/A | 305 | 0.45 | 32768 | 0.4000 | 0.7463 |
| ADAM | N/A | 199 | 0.33 | 50000 | 1.0680 | 0.5750 |
| ADADAGRAD | $\eta = 0.025$ | 222 | 0.32 | 44934 | 1.2770 | 0.5107 |
| ADADAGRAD | $\eta = 0.050$ | 485 | 0.60 | 20615 | 0.6204 | 0.7079 |
| ADADAGRAD | $\eta = 0.075$ | **1697** | **1.02** | **5892** | **0.0258** | **0.8180** |
| ADADAGRAD | $\eta = 0.100$ | **1404** | **0.94** | **7123** | **0.0668** | **0.8085** |
| ADADAM | $\eta = 0.025$ | 211 | 0.34 | 47380 | 0.9039 | 0.6234 |
| ADADAM | $\eta = 0.050$ | 426 | 0.60 | 23463 | 0.0061 | 0.8228 |
| ADADAM | $\eta = 0.075$ | 900 | 0.74 | 11108 | 0.0008 | 0.8983 |
| ADADAM | $\eta = 0.100$ | **1126** | **0.82** | **8880** | **0.0000** | **0.9042** |

gradient optimizers. The computational overhead introduced by the batch-size tests could limit the practical use of the proposed methods in large-scale applications; further engineering effort, as well as development for distributed training, is necessary to fully realize the potential benefits of the proposed adaptive batch size schemes.

# 7    Concluding Remarks

In this work, we demonstrate the versatility of adaptive sampling methods as generic adaptive batch size schemes for adaptive gradient optimizers, supported by both convergence guarantees and numerical results. This opens up several promising research directions for adaptive batch size schemes in large-scale model training. On the theoretical side, it would be interesting to study the convergence guarantees of this class of methods when combined with other stochastic gradient optimizers, such as momentum-based methods, as well as proximal SGD methods for constrained problems with deterministic nonsmooth regularizers. On the practical side, exploring

the implementation of adaptive batch size schemes under various parallelism paradigms for large-scale distributed training—including data, tensor, and pipeline parallelism (Shoeybi et al., 2019; Rajbhandari et al., 2020; Smith et al., 2022; Zhao et al., 2023)—is worthwhile; see, e.g., Lau et al. (2025) for data and model parallelism. This line of work aims to ensure that these schemes are viable for large-scale applications such as the (pre-)training of autoregressive language and image models. Furthermore, examining the impact of adaptive batch size schemes for adaptive gradient methods, in contrast to those designed for SGD, particularly for transformer-based language models in addition to the CNN-based vision tasks discussed in this paper, is important. Unlike the marginal utility of adaptive methods for CNNs and RNNs (Wilson et al., 2017), adaptive gradient methods such as ADAM significantly outperform SGD when optimizing transformers (Zhang et al., 2020b,c; Jiang et al., 2023; Kunstner et al., 2023; Pan and Li, 2023; Ahn et al., 2024).

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

# A   Proofs of Main Text

We provide the omitted proofs of the main text in this section.

## A.1   Preparatory Definitions, Propositions and Lemmas

We give various additional technical definitions, propositions and lemmas before giving the proofs of the theorems.

### A.1.1   Formal Statements Corresponding to Theorem 5

We now state precise versions of Theorem 5.

**Proposition 10** (Exact variance norm test). *Suppose that, for every iteration $k \in \mathbb{N}^*$, the batch gradient $\nabla F_{\mathcal{B}_k}(x_k)$ is conditionally unbiased, that is,*

$$\mathbb{E}_k[\nabla F_{\mathcal{B}_k}(x_k)] = \nabla F(x_k),$$

*and the exact variance norm test (2) holds with some constant $\eta > 0$:*

$$\mathbb{E}_k\big[\|\nabla F_{\mathcal{B}_k}(x_k) - \nabla F(x_k)\|^2\big] \leqslant \eta^2 \|\nabla F(x_k)\|^2.$$

*Then*
$$\mathbb{E}_k\big[\|\nabla F_{\mathcal{B}_k}(x_k)\|^2\big] \leqslant (1 + \eta^2)\|\nabla F(x_k)\|^2.$$

**Proof**  Using the conditional unbiasedness of $\nabla F_{\mathcal{B}_k}(x_k)$, we have

$$\mathbb{E}_k\big[\|\nabla F_{\mathcal{B}_k}(x_k) - \nabla F(x_k)\|^2\big] = \mathbb{E}_k\big[\|\nabla F_{\mathcal{B}_k}(x_k)\|^2\big] - 2\langle \mathbb{E}_k[\nabla F_{\mathcal{B}_k}(x_k)], \nabla F(x_k)\rangle + \|\nabla F(x_k)\|^2$$
$$= \mathbb{E}_k\big[\|\nabla F_{\mathcal{B}_k}(x_k)\|^2\big] - \|\nabla F(x_k)\|^2.$$

Combining this identity with (2) yields the claim. ∎

**Proposition 11** (Exact variance inner product test and orthogonality test). *Suppose that, for every iteration $k \in \mathbb{N}^*$ with $\nabla F(x_k) \neq 0$, the samples in $\mathcal{B}_k$ are conditionally i.i.d., and the exact variance inner product test (5) and exact variance orthogonality test (7) hold with constants $\vartheta > 0$ and $\nu > 0$, respectively. Then*

$$\mathbb{E}_k\big[\|\nabla F_{\mathcal{B}_k}(x_k)\|^2\big] \leqslant (1 + \vartheta^2 + \nu^2)\|\nabla F(x_k)\|^2.$$

**Proof** This is part of the result of Bollapragada et al. (2018a), Lemma 3.1. We include its proof here with our notation for completeness.

Write the batch gradient as

$$\nabla F_{\mathcal{B}_k}(x_k) = \frac{1}{b_k} \sum_{r=1}^{b_k} \nabla f(x_k; \xi_{k,r}),$$

where $\xi_{k,1}, \ldots, \xi_{k,b_k}$ are the conditionally i.i.d. samples in $\mathcal{B}_k$. For $r = 1, \ldots, b_k$, define

$$a_{k,r} := \langle \nabla f(x_k; \xi_{k,r}), \nabla F(x_k) \rangle - \|\nabla F(x_k)\|^2$$

and

$$u_{k,r} := \nabla f(x_k; \xi_{k,r}) - \frac{\langle \nabla f(x_k; \xi_{k,r}), \nabla F(x_k) \rangle}{\|\nabla F(x_k)\|^2} \nabla F(x_k).$$

Since the samples are conditionally i.i.d. and

$$\mathbb{E}_k[\nabla f(x_k; \xi_{k,r})] = \nabla F(x_k),$$

we have $\mathbb{E}_k[a_{k,r}] = 0$ and $\mathbb{E}_k[u_{k,r}] = 0$. Hence

$$\langle \nabla F_{\mathcal{B}_k}(x_k), \nabla F(x_k) \rangle - \|\nabla F(x_k)\|^2 = \frac{1}{b_k} \sum_{r=1}^{b_k} a_{k,r},$$

and

$$\nabla F_{\mathcal{B}_k}(x_k) - \frac{\langle \nabla F_{\mathcal{B}_k}(x_k), \nabla F(x_k) \rangle}{\|\nabla F(x_k)\|^2} \nabla F(x_k) = \frac{1}{b_k} \sum_{r=1}^{b_k} u_{k,r}.$$

Therefore, by conditional independence and centering,

$$
\begin{aligned}
\mathbb{E}_k\left[\left(\langle \nabla F_{\mathcal{B}_k}(x_k), \nabla F(x_k) \rangle - \|\nabla F(x_k)\|^2\right)^2\right] &= \mathbb{E}_k\left[\left(\frac{1}{b_k} \sum_{r=1}^{b_k} a_{k,r}\right)^2\right] \\
&= \frac{1}{b_k^2} \sum_{r=1}^{b_k} \mathbb{E}_k[a_{k,r}^2] \\
&= \frac{1}{b_k} \mathbb{E}_k[a_{k,1}^2] \\
&\leqslant \vartheta^2 \|\nabla F(x_k)\|^4,
\end{aligned}
$$

and similarly

$$\mathbb{E}_k\left[\left\|\nabla F_{\mathcal{B}_k}(x_k) - \frac{\langle\nabla F_{\mathcal{B}_k}(x_k), \nabla F(x_k)\rangle}{\|\nabla F(x_k)\|^2}\nabla F(x_k)\right\|^2\right] = \mathbb{E}_k\left[\left\|\frac{1}{b_k}\sum_{r=1}^{b_k} u_{k,r}\right\|^2\right]$$

$$= \frac{1}{b_k^2}\sum_{r=1}^{b_k}\mathbb{E}_k[\|u_{k,r}\|^2]$$

$$= \frac{1}{b_k}\mathbb{E}_k[\|u_{k,1}\|^2]$$

$$\leqslant \nu^2\|\nabla F(x_k)\|^2.$$

Now decompose $\nabla F_{\mathcal{B}_k}(x_k)$ into its components parallel and orthogonal to $\nabla F(x_k)$:

$$\nabla F_{\mathcal{B}_k}(x_k) = \frac{\langle\nabla F_{\mathcal{B}_k}(x_k), \nabla F(x_k)\rangle}{\|\nabla F(x_k)\|^2}\nabla F(x_k) + r_k,$$

where

$$r_k := \nabla F_{\mathcal{B}_k}(x_k) - \frac{\langle\nabla F_{\mathcal{B}_k}(x_k), \nabla F(x_k)\rangle}{\|\nabla F(x_k)\|^2}\nabla F(x_k).$$

Since the two components are orthogonal,

$$\|\nabla F_{\mathcal{B}_k}(x_k)\|^2 = \frac{(\langle\nabla F_{\mathcal{B}_k}(x_k), \nabla F(x_k)\rangle)^2}{\|\nabla F(x_k)\|^2} + \|r_k\|^2.$$

Taking conditional expectation and using

$$\mathbb{E}_k[\langle\nabla F_{\mathcal{B}_k}(x_k), \nabla F(x_k)\rangle] = \|\nabla F(x_k)\|^2,$$

we obtain

$$\mathbb{E}_k\left[\|\nabla F_{\mathcal{B}_k}(x_k)\|^2\right] = \frac{\mathbb{E}_k\left[(\langle\nabla F_{\mathcal{B}_k}(x_k), \nabla F(x_k)\rangle)^2\right]}{\|\nabla F(x_k)\|^2} + \mathbb{E}_k[\|r_k\|^2]$$

$$= \frac{\mathbb{E}_k\left[(\langle\nabla F_{\mathcal{B}_k}(x_k), \nabla F(x_k)\rangle - \|\nabla F(x_k)\|^2)^2\right] + \|\nabla F(x_k)\|^4}{\|\nabla F(x_k)\|^2} + \mathbb{E}_k[\|r_k\|^2]$$

$$\leqslant (1 + \vartheta^2 + \nu^2)\|\nabla F(x_k)\|^2.$$

$\blacksquare$

### A.1.2   Technical Lemmas

We first record a simple summation lemma for nonnegative sequences.

**Lemma 12.** *Let $(a_k)_{k \in \mathbb{N}} \subset \mathbb{R}_+$ with $a_0 > 0$, and define $s_k := \sum_{i=0}^{k} a_i$ for each $k \in \mathbb{N}$. Then*

$$\sum_{k=1}^{K} \frac{a_k}{s_k^{3/2}} \leqslant \frac{2}{\sqrt{a_0}},$$

$$\sum_{k=1}^{K} \frac{a_k}{s_k} \leqslant \log s_K - \log a_0.$$

**Proof** For each $k \in [\![K]\!]$, we have $a_k = s_k - s_{k-1}$. Since the function $t \mapsto t^{-3/2}$ is decreasing on $\mathbb{R}_{++}$,

$$\frac{a_k}{s_k^{3/2}} = \frac{s_k - s_{k-1}}{s_k^{3/2}} \leqslant \int_{s_{k-1}}^{s_k} t^{-3/2} \, \mathrm{d}t.$$

Summing over $k = 1, \ldots, K$ yields

$$\sum_{k=1}^{K} \frac{a_k}{s_k^{3/2}} \leqslant \int_{a_0}^{s_K} t^{-3/2} \, \mathrm{d}t = \frac{2}{\sqrt{a_0}} - \frac{2}{\sqrt{s_K}} \leqslant \frac{2}{\sqrt{a_0}}.$$

Similarly, since $t \mapsto t^{-1}$ is decreasing on $\mathbb{R}_{++}$,

$$\frac{a_k}{s_k} = \frac{s_k - s_{k-1}}{s_k} \leqslant \int_{s_{k-1}}^{s_k} t^{-1} \, \mathrm{d}t.$$

Summing over $k = 1, \ldots, K$ gives

$$\sum_{k=1}^{K} \frac{a_k}{s_k} \leqslant \int_{a_0}^{s_K} t^{-1} \, \mathrm{d}t = \log s_K - \log a_0.$$

∎

We also state without proof the descent lemma for $(L_0, L_1)$-smooth functions; see Zhang et al. (2020a), Lemma A.3.

**Lemma 13** (Descent lemma for $(L_0, L_1)$-smooth functions)**.** *If Theorem 3 holds, then for any $x, y \in \mathbb{R}^d$ satisfying $L_1 \|x - y\| \leqslant 1$,*

$$F(y) \leqslant F(x) + \langle \nabla F(x), y - x \rangle + \frac{L_0 + L_1 \|\nabla F(x)\|}{2} \|x - y\|^2.$$

## A.2 Convergence Results for ADAGRAD-NORM and ADAGRAD

### A.2.1 Proof of Theorem 6

We start by providing two results.

**Lemma 14.** *Define*

$$\varphi_0 := \frac{\|\nabla F(x_1)\|^2}{\sqrt{v_0}}, \qquad (\forall k \in \mathbb{N}^*) \quad \varphi_k := \frac{\|\nabla F(x_k)\|^2}{\sqrt{v_k}},$$

*and adopt the convention $g_0 := 0$. Then, for any $\rho > 0$ and every $k \in \mathbb{N}^*$,*

$$\left\langle \nabla F(x_k), \mathbb{E}_k\left[\left(\frac{1}{\sqrt{v_{k-1}}} - \frac{1}{\sqrt{v_k}}\right)g_k\right]\right\rangle$$
$$\leqslant \frac{1}{2}\left(1 + \frac{1}{\rho}\right)\frac{\|\nabla F(x_k)\|^2}{\sqrt{v_{k-1}}} + \frac{\tau}{2}\mathbb{E}_k[\varphi_{k-1} - \varphi_k] + \frac{\tau L^2 \alpha^2}{2}(1 + \rho\tau)\frac{\|g_{k-1}\|^2}{v_{k-1}^{3/2}}. \quad (14)$$

**Proof** [Proof of Theorem 14] Note that

$$\frac{1}{\sqrt{v_{k-1}}} - \frac{1}{\sqrt{v_k}} = \frac{v_k - v_{k-1}}{\sqrt{v_{k-1}}\sqrt{v_k}(\sqrt{v_k} + \sqrt{v_{k-1}})} = \frac{\|g_k\|^2}{\sqrt{v_{k-1}}\sqrt{v_k}(\sqrt{v_k} + \sqrt{v_{k-1}})}. \quad (15)$$

Hence

$$\left\langle \nabla F(x_k), \mathbb{E}_k\left[\left(\frac{1}{\sqrt{v_{k-1}}} - \frac{1}{\sqrt{v_k}}\right)g_k\right]\right\rangle$$

$$\leqslant \frac{\|\nabla F(x_k)\|}{\sqrt{v_{k-1}}}\mathbb{E}_k\left[\frac{\|g_k\|^3}{\sqrt{v_k}(\sqrt{v_k} + \sqrt{v_{k-1}})}\right]$$

$$\leqslant \frac{\|\nabla F(x_k)\|}{\sqrt{v_{k-1}}}\mathbb{E}_k\left[\frac{\|g_k\|^2}{\sqrt{v_k} + \sqrt{v_{k-1}}}\right] \qquad \text{since } v_k \geqslant \|g_k\|^2$$

$$\leqslant \frac{\|\nabla F(x_k)\|^2}{2\sqrt{v_{k-1}}} + \frac{1}{2\sqrt{v_{k-1}}}\left(\mathbb{E}_k\left[\frac{\|g_k\|^2}{\sqrt{v_k} + \sqrt{v_{k-1}}}\right]\right)^2$$

$$\leqslant \frac{\|\nabla F(x_k)\|^2}{2\sqrt{v_{k-1}}} + \frac{1}{2\sqrt{v_{k-1}}}\mathbb{E}_k[\|g_k\|^2]\mathbb{E}_k\left[\frac{\|g_k\|^2}{(\sqrt{v_k} + \sqrt{v_{k-1}})^2}\right]$$

$$\leqslant \frac{\|\nabla F(x_k)\|^2}{2\sqrt{v_{k-1}}} + \frac{\tau\|\nabla F(x_k)\|^2}{2\sqrt{v_{k-1}}}\mathbb{E}_k\left[\frac{\|g_k\|^2}{(\sqrt{v_k} + \sqrt{v_{k-1}})^2}\right].$$

Moreover, by (15),

$$\frac{\|g_k\|^2}{\sqrt{v_{k-1}}(\sqrt{v_k} + \sqrt{v_{k-1}})^2} \leqslant \frac{\|g_k\|^2}{\sqrt{v_{k-1}}\sqrt{v_k}(\sqrt{v_k} + \sqrt{v_{k-1}})} = \frac{1}{\sqrt{v_{k-1}}} - \frac{1}{\sqrt{v_k}}.$$

Therefore,

$$\left\langle \nabla F(x_k), \mathbb{E}_k\left[\left(\frac{1}{\sqrt{v_{k-1}}} - \frac{1}{\sqrt{v_k}}\right)g_k\right]\right\rangle \leqslant \frac{\|\nabla F(x_k)\|^2}{2\sqrt{v_{k-1}}} + \frac{\tau}{2}\|\nabla F(x_k)\|^2\mathbb{E}_k\left[\frac{1}{\sqrt{v_{k-1}}} - \frac{1}{\sqrt{v_k}}\right]. \quad (16)$$

For $k = 1$, the claim follows immediately from (16), the definition of $\varphi_0$, the inequality $\frac{1}{2} \leqslant \frac{1}{2}(1 + \rho^{-1})$, and the convention $g_0 = 0$. Assume now that $k \geqslant 2$.

We decompose

$$\|\nabla F(x_k)\|^2\mathbb{E}_k\left[\frac{1}{\sqrt{v_{k-1}}} - \frac{1}{\sqrt{v_k}}\right]$$
$$= \mathbb{E}_k\left[\frac{\|\nabla F(x_{k-1})\|^2}{\sqrt{v_{k-1}}} - \frac{\|\nabla F(x_k)\|^2}{\sqrt{v_k}}\right] + \frac{\|\nabla F(x_k)\|^2 - \|\nabla F(x_{k-1})\|^2}{\sqrt{v_{k-1}}}.$$

By the reverse triangle inequality and Theorem 2,

$$\|\nabla F(x_k)\| - \|\nabla F(x_{k-1})\| \leqslant \|\nabla F(x_k) - \nabla F(x_{k-1})\| \leqslant L\|x_k - x_{k-1}\|,$$

and by the triangle inequality,

$$\|\nabla F(x_{k-1})\| \leqslant \|\nabla F(x_k)\| + L\|x_k - x_{k-1}\|.$$

Hence

$$\|\nabla F(x_k)\|^2 - \|\nabla F(x_{k-1})\|^2 = (\|\nabla F(x_k)\| - \|\nabla F(x_{k-1})\|)(\|\nabla F(x_k)\| + \|\nabla F(x_{k-1})\|)$$
$$\leqslant L^2\|x_k - x_{k-1}\|^2 + 2L\|\nabla F(x_k)\|\|x_k - x_{k-1}\|.$$

Since $x_k - x_{k-1} = -\alpha g_{k-1}/\sqrt{v_{k-1}}$, we obtain

$$\|\nabla F(x_k)\|^2 \mathbb{E}_k\left[\frac{1}{\sqrt{v_{k-1}}} - \frac{1}{\sqrt{v_k}}\right]$$
$$\leqslant \mathbb{E}_k[\varphi_{k-1} - \varphi_k] + \frac{L^2\alpha^2}{v_{k-1}^{3/2}}\|g_{k-1}\|^2 + \frac{2L\alpha}{v_{k-1}}\|\nabla F(x_k)\|\|g_{k-1}\|.$$

Applying Young's inequality with parameter $\rho > 0$,

$$\frac{\tau}{2} \cdot \frac{2L\alpha}{v_{k-1}}\|\nabla F(x_k)\|\|g_{k-1}\| \leqslant \frac{1}{2\rho}\frac{\|\nabla F(x_k)\|^2}{\sqrt{v_{k-1}}} + \frac{\rho\tau^2 L^2\alpha^2}{2}\frac{\|g_{k-1}\|^2}{v_{k-1}^{3/2}}.$$

Substituting this bound into (16) yields

$$\left\langle \nabla F(x_k), \mathbb{E}_k\left[\left(\frac{1}{\sqrt{v_{k-1}}} - \frac{1}{\sqrt{v_k}}\right)g_k\right]\right\rangle$$
$$\leqslant \frac{\|\nabla F(x_k)\|^2}{2\sqrt{v_{k-1}}} + \frac{\tau}{2}\mathbb{E}_k[\varphi_{k-1} - \varphi_k] + \frac{\tau L^2\alpha^2}{2}\frac{\|g_{k-1}\|^2}{v_{k-1}^{3/2}} + \frac{1}{2\rho}\frac{\|\nabla F(x_k)\|^2}{\sqrt{v_{k-1}}} + \frac{\rho\tau^2 L^2\alpha^2}{2}\frac{\|g_{k-1}\|^2}{v_{k-1}^{3/2}}$$
$$= \frac{1}{2}\left(1 + \frac{1}{\rho}\right)\frac{\|\nabla F(x_k)\|^2}{\sqrt{v_{k-1}}} + \frac{\tau}{2}\mathbb{E}_k[\varphi_{k-1} - \varphi_k] + \frac{\tau L^2\alpha^2}{2}(1 + \rho\tau)\frac{\|g_{k-1}\|^2}{v_{k-1}^{3/2}},$$

which completes the proof. ∎

**Lemma 15.** *For any positive constants* $(a, b) \in \mathbb{R}_{++}^2$, *if* $x > 0$ *satisfies* $x \leqslant a + b\log x$, *then* $x \leqslant 2a - 2b + 4b\log(b/2 + 1) \leqslant 2a + 4b\log(b/2 + 1)$.

**Proof** [Proof of Theorem 15] Let $h(t) := t/2 - b\log t$ for $t > 0$. Then

$$h'(t) = 1/2 - b/t, \qquad h''(t) = b/t^2 > 0.$$

Hence $h$ is minimized at $t = 2b$, so

$$h(t) \geqslant h(2b) = b - b\log(2b).$$

Applying this with $t = x$, we obtain

$$x - b \log x = x/2 + (x/2 - b \log x) \geqslant x/2 + b - b \log(2b).$$

Since $x \leqslant a + b \log x$, equivalently $x - b \log x \leqslant a$, it follows that

$$x/2 + b - b \log(2b) \leqslant a,$$

and therefore

$$x \leqslant 2a - 2b + 2b \log(2b).$$

By the A.M.-G.M. inequality, $b + 2 \geqslant 2\sqrt{2b}$, and taking logarithms gives

$$\log(b + 2) \geqslant \log 2 + \frac{1}{2}(\log 2 + \log b).$$

Therefore,

$$b \log(2b) = b(\log 2 + \log b) \leqslant 2b(\log(b + 2) - \log 2) = 2b \log(b/2 + 1).$$

Substituting this bound into the previous inequality yields

$$x \leqslant 2a - 2b + 4b \log(b/2 + 1),$$

and the final inequality is immediate. ∎

If $\nabla F(x_k) = 0$ for some $k \in [\![K]\!]$, then the conclusion is immediate. Hence, when the augmented inner product test is used, we may assume $\nabla F(x_k) \neq 0$ for all $k \in [\![K]\!]$. Under the corresponding hypotheses of Theorems 10 and 11, for every $k \in \mathbb{N}^*$,

$$\mathbb{E}_k[\|g_k\|^2] \leqslant \tau \|\nabla F(x_k)\|^2. \tag{17}$$

By Theorem 2,

$$F(x_{k+1}) \leqslant F(x_k) + \langle \nabla F(x_k), x_{k+1} - x_k \rangle + \frac{L}{2}\|x_{k+1} - x_k\|^2$$

$$= F(x_k) - \alpha \langle \nabla F(x_k), \frac{g_k}{\sqrt{v_k}} \rangle + \frac{L\alpha^2}{2} \frac{\|g_k\|^2}{v_k}.$$

Taking conditional expectation with respect to $\mathcal{F}_k$ gives

$$\mathbb{E}_k[F(x_{k+1})] \leqslant F(x_k) - \alpha \left\langle \nabla F(x_k), \mathbb{E}_k\left[\frac{g_k}{\sqrt{v_k}}\right] \right\rangle + \frac{L\alpha^2}{2} \mathbb{E}_k\left[\frac{\|g_k\|^2}{v_k}\right].$$

Moreover,

$$\left\langle \nabla F(x_k), \mathbb{E}_k\left[\frac{g_k}{\sqrt{v_k}}\right]\right\rangle = \left\langle \nabla F(x_k), \mathbb{E}_k\left[\frac{g_k}{\sqrt{v_{k-1}}}\right]\right\rangle + \left\langle \nabla F(x_k), \mathbb{E}_k\left[\left(\frac{1}{\sqrt{v_k}} - \frac{1}{\sqrt{v_{k-1}}}\right)g_k\right]\right\rangle$$

$$= \frac{\|\nabla F(x_k)\|^2}{\sqrt{v_{k-1}}} + \left\langle \nabla F(x_k), \mathbb{E}_k\left[\left(\frac{1}{\sqrt{v_k}} - \frac{1}{\sqrt{v_{k-1}}}\right)g_k\right]\right\rangle. \qquad (18)$$

Hence

$$\mathbb{E}_k[F(x_{k+1})] \leqslant F(x_k) - \alpha\frac{\|\nabla F(x_k)\|^2}{\sqrt{v_{k-1}}}$$

$$+ \alpha\left\langle \nabla F(x_k), \mathbb{E}_k\left[\left(\frac{1}{\sqrt{v_{k-1}}} - \frac{1}{\sqrt{v_k}}\right)g_k\right]\right\rangle + \frac{L\alpha^2}{2}\mathbb{E}_k\left[\frac{\|g_k\|^2}{v_k}\right]. \qquad (19)$$

Combining (19) with Theorem 14 and taking total expectation, we obtain

$$\mathbb{E}[F(x_{k+1})] \leqslant \mathbb{E}[F(x_k)] - \frac{\alpha}{2}\left(1 - \frac{1}{\rho}\right)\mathbb{E}\left[\frac{\|\nabla F(x_k)\|^2}{\sqrt{v_{k-1}}}\right]$$

$$+ \frac{\tau\alpha}{2}\mathbb{E}[\varphi_{k-1} - \varphi_k] + \frac{L\alpha^2}{2}\mathbb{E}\left[\frac{\|g_k\|^2}{v_k}\right] + \frac{\tau L^2\alpha^3}{2}(1 + \rho\tau)\mathbb{E}\left[\frac{\|g_{k-1}\|^2}{v_{k-1}^{3/2}}\right].$$

Now define

$$A_K := \sum_{k=1}^{K}\mathbb{E}\left[\frac{\|\nabla F(x_k)\|^2}{\sqrt{v_{k-1}}}\right].$$

Summing the previous inequality over $k = 1, \ldots, K$, using $F(x_{K+1}) \geqslant F^\star$, $\varphi_K \geqslant 0$, $g_0 = 0$, and Theorem 12, yields

$$\frac{\alpha}{2}\left(1 - \frac{1}{\rho}\right)A_K$$

$$\leqslant F(x_1) - F^\star + \frac{\tau\alpha}{2}\varphi_0 + \frac{L\alpha^2}{2}(\mathbb{E}[\log v_K] - \log v_0) + \frac{\tau L^2\alpha^3}{2}(1 + \rho\tau)\sum_{k=1}^{K}\mathbb{E}\left[\frac{\|g_{k-1}\|^2}{v_{k-1}^{3/2}}\right]$$

$$\leqslant F(x_1) - F^\star + \frac{\tau\alpha}{2}\frac{\|\nabla F(x_1)\|^2}{\sqrt{v_0}} + \frac{L\alpha^2}{2}(\mathbb{E}[\log v_K] - \log v_0) + \tau L^2\alpha^3(1 + \rho\tau)\frac{1}{\sqrt{v_0}}.$$

Therefore,

$$A_K \leqslant c_1 + \frac{L\alpha}{1 - \rho^{-1}}\mathbb{E}[\log v_K] \leqslant c_1^+ + c_2\mathbb{E}[\log v_K].$$

To control $\mathbb{E}[\sqrt{v_K}]$, observe that

$$\sqrt{v_K} = \sqrt{v_0} + \sum_{k=1}^{K}\frac{\|g_k\|^2}{\sqrt{v_k} + \sqrt{v_{k-1}}} \leqslant \sqrt{v_0} + \sum_{k=1}^{K}\frac{\|g_k\|^2}{\sqrt{v_{k-1}}}.$$

Taking expectation and using (17), we get

$$x := \mathbb{E}[\sqrt{v_K}] \leqslant \sqrt{v_0} + \tau A_K \leqslant \sqrt{v_0} + \tau c_1^+ + \tau c_2\mathbb{E}[\log v_K].$$

Since $\log v_K = 2 \log \sqrt{v_K}$ and $\log$ is concave,

$$\mathbb{E}[\log v_K] \leqslant 2 \log \mathbb{E}[\sqrt{v_K}] = 2 \log x.$$

Thus

$$x \leqslant \sqrt{v_0} + \tau c_1^+ + 2\tau c_2 \log x.$$

Applying Theorem 15 with $a = \sqrt{v_0} + \tau c_1^+$ and $b = 2\tau c_2$, we obtain

$$\mathbb{E}[\sqrt{v_K}] \leqslant c_3.$$

Consequently,

$$A_K \leqslant c_1^+ + 2c_2 \log c_3.$$

Now set

$$S_K := \sum_{k=1}^{K} \|\nabla F(x_k)\|^2.$$

Since $v_{k-1} \leqslant v_K$ for all $k \in [\![K]\!]$,

$$A_K \geqslant \mathbb{E}\left[\frac{S_K}{\sqrt{v_K}}\right].$$

By Cauchy-Schwarz,

$$\mathbb{E}[\sqrt{S_K}]^2 \leqslant \mathbb{E}\left[\frac{S_K}{\sqrt{v_K}}\right]\mathbb{E}[\sqrt{v_K}] \leqslant A_K \mathbb{E}[\sqrt{v_K}] \leqslant c_3(c_1^+ + 2c_2 \log c_3).$$

Therefore, for any $\varepsilon > 0$,

$$\mathbb{P}\left(\min_{k \in [\![K]\!]} \|\nabla F(x_k)\|^2 > \varepsilon\right) \leqslant \mathbb{P}(S_K > K\varepsilon) = \mathbb{P}(\sqrt{S_K} > \sqrt{K\varepsilon}) \leqslant \frac{\mathbb{E}[\sqrt{S_K}]}{\sqrt{K\varepsilon}}.$$

Choosing

$$\varepsilon = \frac{c_3(c_1^+ + 2c_2 \log c_3)}{K\delta^2}$$

gives

$$\mathbb{P}\left(\min_{k \in [\![K]\!]} \|\nabla F(x_k)\|^2 > \frac{c_3(c_1^+ + 2c_2 \log c_3)}{K\delta^2}\right) \leqslant \delta,$$

which completes the proof.

### A.2.2 Proof of Theorem 8

The proof follows the same strategy as that of Theorem 6, but the estimates must be carried out coordinate-wise. Write $g_{k,j} := [g_k]_j$, $v_{k,j} := [v_k]_j$, and set

$$\tau := 1 + \eta^2, \qquad H_k := \left\|\frac{1}{\sqrt{v_k}} \odot g_k\right\|^2, \qquad \Gamma_k := \sum_{j=1}^{d} \frac{1}{\sqrt{v_{k,j}}},$$

$$\widetilde{\varphi}_{0,j} := \frac{(\partial_j F(x_1))^2}{\sqrt{v_{0,j}}}, \qquad (\forall k \in \mathbb{N}^*) \quad \widetilde{\varphi}_{k,j} := \frac{(\partial_j F(x_k))^2}{\sqrt{v_{k,j}}}, \qquad j \in [\![d]\!].$$

We also set $H_0 := 0$.

Since $g_k = \nabla F_{\mathcal{B}_k}(x_k)$ is conditionally unbiased, the coordinate-wise exact variance norm test (11) implies

$$\mathbb{E}_k[g_{k,j}^2] = \mathbb{E}_k[(g_{k,j} - \partial_j F(x_k))^2] + (\partial_j F(x_k))^2 \leqslant \tau (\partial_j F(x_k))^2$$

for every $(k,j) \in \mathbb{N}^* \times [\![d]\!]$.

By Theorem 2 and the AdaGrad update $x_{k+1} = x_k - \alpha g_k \odot v_k^{-1/2}$,

$$\mathbb{E}_k[F(x_{k+1})] \leqslant F(x_k) - \alpha \left\langle \nabla F(x_k), \mathbb{E}_k\left[\frac{1}{\sqrt{v_k}} \odot g_k\right] \right\rangle + \frac{L\alpha^2}{2}\mathbb{E}_k[H_k]$$

$$= F(x_k) - \alpha \sum_{j=1}^{d} \frac{(\partial_j F(x_k))^2}{\sqrt{v_{k-1,j}}} + \alpha \sum_{j=1}^{d} T_{k,j} + \frac{L\alpha^2}{2}\mathbb{E}_k[H_k],$$

where

$$T_{k,j} := \partial_j F(x_k)\, \mathbb{E}_k\left[\left(\frac{1}{\sqrt{v_{k-1,j}}} - \frac{1}{\sqrt{v_{k,j}}}\right)g_{k,j}\right].$$

Since $v_{k,j} = v_{k-1,j} + g_{k,j}^2$,

$$\frac{1}{\sqrt{v_{k-1,j}}} - \frac{1}{\sqrt{v_{k,j}}} = \frac{g_{k,j}^2}{\sqrt{v_{k-1,j}}\sqrt{v_{k,j}}(\sqrt{v_{k,j}} + \sqrt{v_{k-1,j}})}.$$

Using $\sqrt{v_{k,j}} \geqslant |g_{k,j}|$, Young's inequality, and Cauchy-Schwarz, we obtain

$$T_{k,j} \leqslant \frac{|\partial_j F(x_k)|}{\sqrt{v_{k-1,j}}}\mathbb{E}_k\left[\frac{|g_{k,j}|^3}{\sqrt{v_{k,j}}(\sqrt{v_{k,j}} + \sqrt{v_{k-1,j}})}\right]$$

$$\leqslant \frac{|\partial_j F(x_k)|}{\sqrt{v_{k-1,j}}}\mathbb{E}_k\left[\frac{g_{k,j}^2}{\sqrt{v_{k,j}} + \sqrt{v_{k-1,j}}}\right]$$

$$\leqslant \frac{(\partial_j F(x_k))^2}{2\sqrt{v_{k-1,j}}} + \frac{1}{2\sqrt{v_{k-1,j}}}\left(\mathbb{E}_k\left[\frac{g_{k,j}^2}{\sqrt{v_{k,j}} + \sqrt{v_{k-1,j}}}\right]\right)^2$$

$$\leqslant \frac{(\partial_j F(x_k))^2}{2\sqrt{v_{k-1,j}}} + \frac{\mathbb{E}_k[g_{k,j}^2]}{2\sqrt{v_{k-1,j}}}\mathbb{E}_k\left[\frac{g_{k,j}^2}{(\sqrt{v_{k,j}} + \sqrt{v_{k-1,j}})^2}\right]$$

$$\leqslant \frac{(\partial_j F(x_k))^2}{2\sqrt{v_{k-1,j}}} + \frac{\tau}{2}(\partial_j F(x_k))^2\mathbb{E}_k\left[\frac{1}{\sqrt{v_{k-1,j}}} - \frac{1}{\sqrt{v_{k,j}}}\right].$$

For $k = 1$, the last term is $\mathbb{E}_1[\widetilde{\varphi}_{0,j} - \widetilde{\varphi}_{1,j}]$. For $k \geqslant 2$,

$$(\partial_j F(x_k))^2\mathbb{E}_k\left[\frac{1}{\sqrt{v_{k-1,j}}} - \frac{1}{\sqrt{v_{k,j}}}\right] = \mathbb{E}_k[\widetilde{\varphi}_{k-1,j} - \widetilde{\varphi}_{k,j}] + \frac{(\partial_j F(x_k))^2 - (\partial_j F(x_{k-1}))^2}{\sqrt{v_{k-1,j}}}.$$

Moreover,

$$|\partial_j F(x_k) - \partial_j F(x_{k-1})| \leqslant \|\nabla F(x_k) - \nabla F(x_{k-1})\| \leqslant L\|x_k - x_{k-1}\| = \alpha L\sqrt{H_{k-1}},$$

so

$$(\partial_j F(x_k))^2 - (\partial_j F(x_{k-1}))^2 \leqslant L^2\alpha^2 H_{k-1} + 2\alpha L|\partial_j F(x_k)|\sqrt{H_{k-1}}.$$

Applying Young's inequality with parameter $\rho$ yields

$$\tau \alpha L \frac{|\partial_j F(x_k)|}{\sqrt{v_{k-1,j}}} \sqrt{H_{k-1}} \leqslant \frac{1}{2\rho} \frac{(\partial_j F(x_k))^2}{\sqrt{v_{k-1,j}}} + \frac{\rho \tau^2 L^2 \alpha^2}{2} \frac{H_{k-1}}{\sqrt{v_{k-1,j}}}.$$

Therefore, for every $k \in [\![K]\!]$ and $j \in [\![d]\!]$,

$$T_{k,j} \leqslant \frac{1}{2}\left(1 + \frac{1}{\rho}\right) \frac{(\partial_j F(x_k))^2}{\sqrt{v_{k-1,j}}} + \frac{\tau}{2} \mathbb{E}_k[\widetilde{\varphi}_{k-1,j} - \widetilde{\varphi}_{k,j}] + \frac{\tau L^2 \alpha^2}{2}(1 + \rho\tau)\frac{H_{k-1}}{\sqrt{v_{k-1,j}}}.$$

Substituting this into the descent inequality, taking full expectation, and summing over $k = 1, \ldots, K$, we obtain

$$\frac{\alpha}{2}\left(1 - \frac{1}{\rho}\right) \sum_{k=1}^{K} \sum_{j=1}^{d} \mathbb{E}\left[\frac{(\partial_j F(x_k))^2}{\sqrt{v_{k-1,j}}}\right]$$
$$\leqslant F(x_1) - F^\star + \frac{\tau\alpha}{2} \sum_{j=1}^{d} \widetilde{\varphi}_{0,j} + \frac{\tau L^2 \alpha^3}{2}(1 + \rho\tau) \sum_{k=1}^{K} \mathbb{E}[\Gamma_{k-1} H_{k-1}] + \frac{L\alpha^2}{2} \sum_{k=1}^{K} \mathbb{E}[H_k].$$

Since $v_{k,j}$ is nondecreasing in $k$, $\Gamma_{k-1} \leqslant \Gamma_0$ for all $k$.

By Theorem 12, for each $j \in [\![d]\!]$,

$$\sum_{k=1}^{K} \frac{g_{k,j}^2}{v_{k,j}} \leqslant \log v_{K,j} - \log v_{0,j}.$$

Hence

$$\sum_{k=1}^{K} \mathbb{E}[H_k] = \sum_{j=1}^{d} \sum_{k=1}^{K} \mathbb{E}\left[\frac{g_{k,j}^2}{v_{k,j}}\right] \leqslant \sum_{j=1}^{d} \mathbb{E}[\log v_{K,j}] - \sum_{j=1}^{d} \log v_{0,j},$$

and

$$\sum_{k=1}^{K} \mathbb{E}[\Gamma_{k-1} H_{k-1}] \leqslant \Gamma_0 \sum_{k=1}^{K-1} \mathbb{E}[H_k] \leqslant \Gamma_0 \sum_{k=1}^{K} \mathbb{E}[H_k].$$

Therefore there exist constants $c_1, c_2 > 0$, independent of $K$ and $\delta$, such that

$$S_K := \sum_{k=1}^{K} \sum_{j=1}^{d} \mathbb{E}\left[\frac{(\partial_j F(x_k))^2}{\sqrt{v_{k-1,j}}}\right] \leqslant c_1 + c_2 \sum_{j=1}^{d} \mathbb{E}[\log v_{K,j}].$$

Now set

$$Y_K := \sum_{j=1}^{d} \sqrt{v_{K,j}}, \qquad y_K := \mathbb{E}[Y_K].$$

Using the coordinate-wise second-moment bound,

$$S_K \geqslant \frac{1}{\tau} \sum_{k=1}^{K} \sum_{j=1}^{d} \mathbb{E}\left[\frac{g_{k,j}^2}{\sqrt{v_{k-1,j}}}\right].$$

For each fixed $j$,

$$\frac{g_{k,j}^2}{\sqrt{v_{k-1,j}}} = \frac{v_{k,j} - v_{k-1,j}}{\sqrt{v_{k-1,j}}} \geqslant 2(\sqrt{v_{k,j}} - \sqrt{v_{k-1,j}}),$$

and hence

$$S_K \geqslant \frac{2}{\tau}\left(y_K - \sum_{j=1}^d \sqrt{v_{0,j}}\right).$$

On the other hand, pointwise,

$$\sum_{j=1}^d \log v_{K,j} \leqslant 2d\log(1 + Y_K).$$

Taking expectation and using Jensen's inequality gives

$$\sum_{j=1}^d \mathbb{E}[\log v_{K,j}] \leqslant 2d\log(1 + y_K).$$

Combining the last three displays yields

$$y_K \leqslant c_3 + c_4 \log(1 + y_K)$$

for some constants $c_3, c_4 > 0$ independent of $K$ and $\delta$. Applying Theorem 15 to $1 + y_K$ shows that $y_K \leqslant c_5$ for some constant $c_5 > 0$ independent of $K$ and $\delta$. Consequently,

$$S_K \leqslant c_6$$

for some constant $c_6 > 0$ independent of $K$ and $\delta$.

Finally, since $v_{k-1,j} \leqslant v_{K,j}$, we have pointwise

$$\sum_{j=1}^d \frac{(\partial_j F(x_k))^2}{\sqrt{v_{k-1,j}}} \geqslant \frac{\|\nabla F(x_k)\|^2}{Y_K}.$$

Therefore,

$$S_K \geqslant \mathbb{E}\left[\frac{1}{Y_K}\sum_{k=1}^K \|\nabla F(x_k)\|^2\right].$$

By Cauchy-Schwarz,

$$\mathbb{E}\left[\sqrt{\sum_{k=1}^K \|\nabla F(x_k)\|^2}\right]^2 \leqslant \mathbb{E}[Y_K]\,\mathbb{E}\left[\frac{1}{Y_K}\sum_{k=1}^K \|\nabla F(x_k)\|^2\right] \leqslant y_K S_K \leqslant c_5 c_6.$$

Set $C := c_5 c_6$. Then, by Markov's inequality,

$$\mathbb{P}\left(\min_{k\in[\![K]\!]} \|\nabla F(x_k)\|^2 > \frac{C}{K\delta^2}\right) \leqslant \mathbb{P}\left(\sqrt{\sum_{k=1}^K \|\nabla F(x_k)\|^2} > \frac{\sqrt{C}}{\delta}\right) \leqslant \delta.$$

Thus, with probability at least $1 - \delta$,

$$\min_{k \in [\![K]\!]} \|\nabla F(x_k)\|^2 \leqslant \frac{C}{K\delta^2},$$

which proves the claim.

### A.2.3 Proof of Theorem 9

In case (i), conditional unbiasedness is assumed. In case (ii), $g_k$ is the average of conditionally i.i.d. per-sample gradients with conditional mean $\nabla F(x_k)$, so it is conditionally unbiased as well. Hence, in both cases,

$$\mathbb{E}_k[g_k] = \nabla F(x_k), \qquad \mathbb{E}_k[\|g_k\|^2] \leqslant \tau\|\nabla F(x_k)\|^2, \qquad k \in [\![K]\!],$$

where the second inequality follows from Theorems 10 and 11.

Set $x_0 := x_1$, $g_0 := 0$, $\Delta_k := x_k - x_{k-1}$ for $k \in \mathbb{N}^*$, and

$$(\forall k \in \mathbb{N}) \qquad \varphi_k := \frac{\|\nabla F(x_k)\|^2}{\sqrt{v_k}}.$$

Since $1/\rho_1 + \rho_1/\rho_2 + 2\omega < 1$, we have $\rho_1 > 1$ and $\omega < 1/2$. As $\tau \geqslant 1$, the step-size assumption implies $\alpha \leqslant 1/L_1$. Moreover,

$$\|x_{k+1} - x_k\| = \alpha\left\|\frac{g_k}{\sqrt{v_k}}\right\| \leqslant \alpha \leqslant \frac{1}{L_1},$$

because $v_k = v_{k-1} + \|g_k\|^2 \geqslant \|g_k\|^2$. Hence Theorem 13 gives, pathwise,

$$F(x_{k+1}) \leqslant F(x_k) - \alpha\left\langle \nabla F(x_k), \frac{g_k}{\sqrt{v_k}}\right\rangle + \frac{\alpha^2}{2}(L_0 + L_1\|\nabla F(x_k)\|)\left\|\frac{g_k}{\sqrt{v_k}}\right\|^2.$$

Taking conditional expectation yields

$$\mathbb{E}_k[F(x_{k+1})] \leqslant F(x_k) - \alpha\left\langle \nabla F(x_k), \mathbb{E}_k\left[\frac{g_k}{\sqrt{v_k}}\right]\right\rangle + \frac{L_0\alpha^2}{2}\mathbb{E}_k\left[\frac{\|g_k\|^2}{v_k}\right] + \frac{L_1\alpha^2\|\nabla F(x_k)\|}{2}\mathbb{E}_k\left[\frac{\|g_k\|^2}{v_k}\right]. \tag{20}$$

Using

$$\left\langle \nabla F(x_k), \mathbb{E}_k\left[\frac{g_k}{\sqrt{v_k}}\right]\right\rangle = \frac{\|\nabla F(x_k)\|^2}{\sqrt{v_{k-1}}} + \left\langle \nabla F(x_k), \mathbb{E}_k\left[\left(\frac{1}{\sqrt{v_k}} - \frac{1}{\sqrt{v_{k-1}}}\right)g_k\right]\right\rangle,$$

we obtain

$$-\alpha\left\langle \nabla F(x_k), \mathbb{E}_k\left[\frac{g_k}{\sqrt{v_k}}\right]\right\rangle = -\alpha\frac{\|\nabla F(x_k)\|^2}{\sqrt{v_{k-1}}} + \alpha\left\langle \nabla F(x_k), \mathbb{E}_k\left[\left(\frac{1}{\sqrt{v_{k-1}}} - \frac{1}{\sqrt{v_k}}\right)g_k\right]\right\rangle.$$

Define

$$Q_k := \frac{\|\nabla F(x_k)\|}{\sqrt{v_{k-1}}}\mathbb{E}_k\left[\frac{\|g_k\|^2}{\sqrt{v_k} + \sqrt{v_{k-1}}}\right].$$

By the identity

$$\frac{1}{\sqrt{v_{k-1}}} - \frac{1}{\sqrt{v_k}} = \frac{\|g_k\|^2}{\sqrt{v_{k-1}}\sqrt{v_k}(\sqrt{v_k} + \sqrt{v_{k-1}})},$$

together with $v_k \geqslant \|g_k\|^2$, we have

$$\left\langle \nabla F(x_k), \mathbb{E}_k\left[\left(\frac{1}{\sqrt{v_{k-1}}} - \frac{1}{\sqrt{v_k}}\right)g_k\right]\right\rangle \leqslant Q_k.$$

Also, since $v_k \geqslant v_{k-1}$,

$$\frac{1}{2v_k} \leqslant \frac{1}{\sqrt{v_{k-1}}(\sqrt{v_k} + \sqrt{v_{k-1}})},$$

and therefore

$$\frac{L_1\alpha^2\|\nabla F(x_k)\|}{2}\mathbb{E}_k\left[\frac{\|g_k\|^2}{v_k}\right] \leqslant L_1\alpha^2 Q_k.$$

Since $\alpha \leqslant 1/L_1$, (20) becomes

$$\mathbb{E}_k[F(x_{k+1})] \leqslant F(x_k) - \alpha\frac{\|\nabla F(x_k)\|^2}{\sqrt{v_{k-1}}} + 2\alpha Q_k + \frac{L_0\alpha^2}{2}\mathbb{E}_k\left[\frac{\|g_k\|^2}{v_k}\right]. \tag{21}$$

Next, by Young's inequality, Cauchy–Schwarz, the E-SG bound, and the inequality

$$\frac{\|g_k\|^2}{\sqrt{v_{k-1}}(\sqrt{v_k} + \sqrt{v_{k-1}})^2} \leqslant \frac{1}{\sqrt{v_{k-1}}} - \frac{1}{\sqrt{v_k}},$$

we get

$$\begin{aligned}
2\alpha Q_k &\leqslant \frac{\alpha}{\rho_1}\frac{\|\nabla F(x_k)\|^2}{\sqrt{v_{k-1}}} + \frac{\rho_1\alpha}{\sqrt{v_{k-1}}}\left(\mathbb{E}_k\left[\frac{\|g_k\|^2}{\sqrt{v_k} + \sqrt{v_{k-1}}}\right]\right)^2 \\
&\leqslant \frac{\alpha}{\rho_1}\frac{\|\nabla F(x_k)\|^2}{\sqrt{v_{k-1}}} + \frac{\rho_1\alpha}{\sqrt{v_{k-1}}}\mathbb{E}_k[\|g_k\|^2]\,\mathbb{E}_k\left[\frac{\|g_k\|^2}{(\sqrt{v_k} + \sqrt{v_{k-1}})^2}\right] \\
&\leqslant \frac{\alpha}{\rho_1}\frac{\|\nabla F(x_k)\|^2}{\sqrt{v_{k-1}}} + \rho_1\alpha\tau\|\nabla F(x_k)\|^2\mathbb{E}_k\left[\frac{1}{\sqrt{v_{k-1}}} - \frac{1}{\sqrt{v_k}}\right].
\end{aligned}$$

We decompose the last term as

$$\|\nabla F(x_k)\|^2\mathbb{E}_k\left[\frac{1}{\sqrt{v_{k-1}}} - \frac{1}{\sqrt{v_k}}\right] = \mathbb{E}_k[\varphi_{k-1} - \varphi_k] + \frac{\|\nabla F(x_k)\|^2 - \|\nabla F(x_{k-1})\|^2}{\sqrt{v_{k-1}}}.$$

By Theorem 3,

$$\|\nabla F(x_k) - \nabla F(x_{k-1})\| \leqslant (L_0 + L_1\|\nabla F(x_k)\|)\|\Delta_k\|,$$

hence, by the triangle inequality and the reverse triangle inequality,

$$\|\nabla F(x_k)\|^2 - \|\nabla F(x_{k-1})\|^2 \leqslant (L_0 + L_1\|\nabla F(x_k)\|)^2\|\Delta_k\|^2 + 2(L_0 + L_1\|\nabla F(x_k)\|)\|\nabla F(x_k)\|\|\Delta_k\|.$$

Using $(a + b)^2 \leqslant 2(a^2 + b^2)$, we obtain

$$\rho_1 \alpha \tau \frac{\|\nabla F(x_k)\|^2 - \|\nabla F(x_{k-1})\|^2}{\sqrt{v_{k-1}}}$$
$$\leqslant \rho_1 \alpha \tau \frac{2L_0^2 \|\Delta_k\|^2 + 2L_1^2 \|\nabla F(x_k)\|^2 \|\Delta_k\|^2 + 2L_0 \|\nabla F(x_k)\| \|\Delta_k\| + 2L_1 \|\nabla F(x_k)\|^2 \|\Delta_k\|}{\sqrt{v_{k-1}}}.$$

Using

$$2\tau L_0 \|\nabla F(x_k)\| \|\Delta_k\| \leqslant \frac{1}{\rho_2} \|\nabla F(x_k)\|^2 + \rho_2 \tau^2 L_0^2 \|\Delta_k\|^2,$$

together with $\|\Delta_k\| \leqslant \alpha$ and the step-size restriction

$$2\rho_1 \tau L_1 \|\Delta_k\| \leqslant \omega, \qquad 2\rho_1 \tau L_1^2 \|\Delta_k\|^2 \leqslant \omega,$$

we obtain

$$\rho_1 \alpha \tau \frac{\|\nabla F(x_k)\|^2 - \|\nabla F(x_{k-1})\|^2}{\sqrt{v_{k-1}}} \leqslant \alpha \left( \frac{\rho_1}{\rho_2} + 2\omega \right) \frac{\|\nabla F(x_k)\|^2}{\sqrt{v_{k-1}}} + \rho_1 L_0^2 \tau (2 + \rho_2 \tau) \alpha \frac{\|\Delta_k\|^2}{\sqrt{v_{k-1}}}.$$

Since $\Delta_k = -\alpha g_{k-1} / \sqrt{v_{k-1}}$, this yields

$$2\alpha Q_k \leqslant \alpha \left( \frac{1}{\rho_1} + \frac{\rho_1}{\rho_2} + 2\omega \right) \frac{\|\nabla F(x_k)\|^2}{\sqrt{v_{k-1}}} + \rho_1 \alpha \tau \mathbb{E}_k[\varphi_{k-1} - \varphi_k]$$
$$+ \rho_1 L_0^2 \tau (2 + \rho_2 \tau) \alpha^3 \frac{\|g_{k-1}\|^2}{v_{k-1}^{3/2}}. \quad (22)$$

Combining (21) and (22), and setting

$$c := 1 - \frac{1}{\rho_1} - \frac{\rho_1}{\rho_2} - 2\omega > 0,$$

we obtain

$$\mathbb{E}_k[F(x_{k+1})] \leqslant F(x_k) - c\alpha \frac{\|\nabla F(x_k)\|^2}{\sqrt{v_{k-1}}} + \rho_1 \alpha \tau \mathbb{E}_k[\varphi_{k-1} - \varphi_k]$$
$$+ \frac{L_0 \alpha^2}{2} \mathbb{E}_k \left[ \frac{\|g_k\|^2}{v_k} \right] + \rho_1 L_0^2 \tau (2 + \rho_2 \tau) \alpha^3 \frac{\|g_{k-1}\|^2}{v_{k-1}^{3/2}}. \quad (23)$$

Taking total expectation and summing (23) over $k = 1, \ldots, K$, then using $F(x_{K+1}) \geqslant F^\star$, $\varphi_K \geqslant 0$, and Theorem 12, gives

$$\sum_{k=1}^{K} \mathbb{E} \left[ \frac{\|\nabla F(x_k)\|^2}{\sqrt{v_{k-1}}} \right] \leqslant A + B \, \mathbb{E}[\log v_K]$$

for some finite constants $A, B > 0$ independent of $K$ and $\delta$. Indeed,

$$\sum_{k=1}^{K} \frac{\|g_k\|^2}{v_k} \leqslant \log v_K - \log v_0, \qquad \sum_{k=1}^{K} \frac{\|g_{k-1}\|^2}{v_{k-1}^{3/2}} \leqslant \frac{2}{\sqrt{v_0}},$$

and $\sum_{k=1}^{K} \mathbb{E}[\varphi_{k-1} - \varphi_k] = \mathbb{E}[\varphi_0 - \varphi_K] \leqslant \varphi_0$.

Next,

$$\mathbb{E}[\sqrt{v_K}] = \sqrt{v_0} + \sum_{k=1}^{K} \mathbb{E}\left[\frac{\|g_k\|^2}{\sqrt{v_k} + \sqrt{v_{k-1}}}\right]$$

$$\leqslant \sqrt{v_0} + \frac{1}{2}\sum_{k=1}^{K} \mathbb{E}\left[\frac{\|g_k\|^2}{\sqrt{v_{k-1}}}\right]$$

$$\leqslant \sqrt{v_0} + \frac{\tau}{2}\sum_{k=1}^{K} \mathbb{E}\left[\frac{\|\nabla F(x_k)\|^2}{\sqrt{v_{k-1}}}\right]$$

$$\leqslant \sqrt{v_0} + \frac{\tau}{2}(A + B\,\mathbb{E}[\log v_K]).$$

Since $\mathbb{E}[\log v_K] = 2\mathbb{E}[\log\sqrt{v_K}] \leqslant 2\log\mathbb{E}[\sqrt{v_K}]$ by Jensen's inequality, Theorem 15 implies that $\mathbb{E}[\sqrt{v_K}] \leqslant C_v$ for some finite constant $C_v > 0$ independent of $K$. Consequently,

$$\sum_{k=1}^{K} \mathbb{E}\left[\frac{\|\nabla F(x_k)\|^2}{\sqrt{v_{k-1}}}\right] \leqslant A + 2B\log C_v =: C_s.$$

Since $v_{k-1} \leqslant v_K$, we have

$$\sum_{k=1}^{K} \mathbb{E}\left[\frac{\|\nabla F(x_k)\|^2}{\sqrt{v_{k-1}}}\right] \geqslant \mathbb{E}\left[\frac{1}{\sqrt{v_K}}\sum_{k=1}^{K} \|\nabla F(x_k)\|^2\right].$$

By Cauchy–Schwarz,

$$\mathbb{E}\left[\sqrt{\sum_{k=1}^{K} \|\nabla F(x_k)\|^2}\right]^2 \leqslant \mathbb{E}[\sqrt{v_K}]\,\mathbb{E}\left[\frac{1}{\sqrt{v_K}}\sum_{k=1}^{K} \|\nabla F(x_k)\|^2\right] \leqslant C_v C_s.$$

Finally, since

$$\sum_{k=1}^{K} \|\nabla F(x_k)\|^2 \geqslant K \min_{k\in[\![K]\!]} \|\nabla F(x_k)\|^2,$$

Markov's inequality yields

$$\mathbb{P}\left(\min_{k\in[\![K]\!]} \|\nabla F(x_k)\|^2 > \frac{C_v C_s}{K\delta^2}\right) \leqslant \mathbb{P}\left(\sqrt{\sum_{k=1}^{K} \|\nabla F(x_k)\|^2} > \frac{\sqrt{C_v C_s}}{\delta}\right) \leqslant \delta.$$

Therefore, with probability at least $1 - \delta$,

$$\min_{k\in[\![K]\!]} \|\nabla F(x_k)\|^2 \leqslant \frac{C_v C_s}{K\delta^2}.$$

This proves the claim with $C := C_v C_s$.

# B  Details and Additional Results of Numerical Experiments

In this section, we provide details and additional results for the numerical experiments in Section 6. In particular, we report the training hyperparameters used in the experiments and include additional plots not shown in the main text.

**Sampling protocol, test frequency, and numerical safeguards.**  All experiments use the standard shuffled-epoch sampling protocol. At the beginning of each epoch, the training data are randomly shuffled, and minibatches are then processed sequentially from the shuffled data order. Thus, the empirical implementation follows the usual without-replacement shuffled-epoch protocol rather than independent sampling with replacement. In the adaptive-batch experiments, the batch-size statistic computed from the current minibatch is used to determine the batch size for the next adaptive update. The adaptive test is evaluated once per optimizer iteration during the adaptive phase, except for the three-layer CNN on CIFAR-10 experiment, where it is evaluated every 10 optimizer iterations. Once the prescribed maximum batch size is reached, we stop evaluating the adaptive test and continue training with the capped batch size. The initial (base) batch sizes and maximum batch sizes for all experiments are reported in their corresponding training hyperparameter tables. We did not apply denominator clipping or any other special numerical treatment to potentially small empirical gradient norms appearing in the denominators of the batch-size test statistics. Empirically, the maximum batch size was typically reached at an early or intermediate stage of training, so near-stationary regimes with very small empirical gradient norms did not require denominator clipping or regularization in our reported experiments.

**Aggregate cost reporting.**  To make the computational comparison more transparent, we report aggregate cost statistics for the experimental runs in individual subsections. The reported wall-clock time is the end-to-end training time and includes the cost of computing the empirical adaptive-test statistics, including the per-sample gradient quantities used by the tests. However, our implementation does not separately instrument the forward/backward pass, the per-sample-gradient-statistic computation, the batch-size-test computation, or GPU memory usage. Thus, the reported timings should be interpreted as aggregate cost measurements rather than as an isolated decomposition of adaptive-test overhead or memory overhead.

## B.1  Multi-class Logistic Regression on MNIST

The following table lists the training specifications and optimizer hyperparameters for the experiments on multi-class logistic regression on the MNIST dataset.

Table 7 provides aggregate cost statistics for the experiments on multi-class logistic regression on the MNIST dataset, and we also plot the training loss curves against the number of iterations in Figure 6.

Table 6: Training hyperparameters for multi-class logistic regression on MNIST

| Model | Multi-class Logistic Regression |
|---|---|
| Training budget | 6,000,000 samples (100 epochs) |
| Weight initialization | Default |
| Learning rate schedule | None |
| Optimizer | SGD or ADAGRAD(-NORM) |
| Base learning rate | 0.008 |
| Base batch size | 2 |
| Maximum global batch size | 60,000 |
| Weight decay | 0 |
| Momentum | 0 |
| Precision | tf32 |

Table 7: Aggregate cost statistics for multi-class logistic regression on MNIST

| Scheme | test | time (h) | samples | steps | batch size | loss | accuracy |
|---|---|---|---|---|---|---|---|
| ADASGD | $\eta = 0.10$ | 0.40 | 6,030,188 | 351 | 17131 | 1.04 | 0.82 |
| ADASGD | $\eta = 0.25$ | 0.91 | 6,006,252 | 1029 | 5831 | 0.67 | 0.86 |
| ADADAGRAD-NORM | $\eta = 0.10$ | 0.63 | 6,005,933 | 596 | 10060 | 1.50 | 0.78 |
| ADADAGRAD-NORM | $\eta = 0.25$ | 1.91 | 6,001,106 | 2462 | 2437 | 1.02 | 0.83 |
| ADADAGRAD | $\eta = 0.10$ | **0.15** | **6,004,856** | **126** | **47282** | **0.54** | **0.88** |
| ADADAGRAD | $\eta = 0.25$ | **0.30** | **6,027,458** | **274** | **21918** | **0.46** | **0.90** |
| ADASGD | $\vartheta = 0.01$ | 1.11 | 6,003,750 | 717 | 8362 | 0.76 | 0.85 |
| ADASGD | $\vartheta = 0.05$ | 2.13 | 6,005,454 | 1804 | 3327 | 0.54 | 0.88 |
| ADADAGRAD-NORM | $\vartheta = 0.01$ | 1.50 | 6,002,839 | 1127 | 5322 | 1.28 | 0.80 |
| ADADAGRAD-NORM | $\vartheta = 0.05$ | 5.37 | 6,001,053 | 5349 | 1122 | 0.80 | 0.85 |

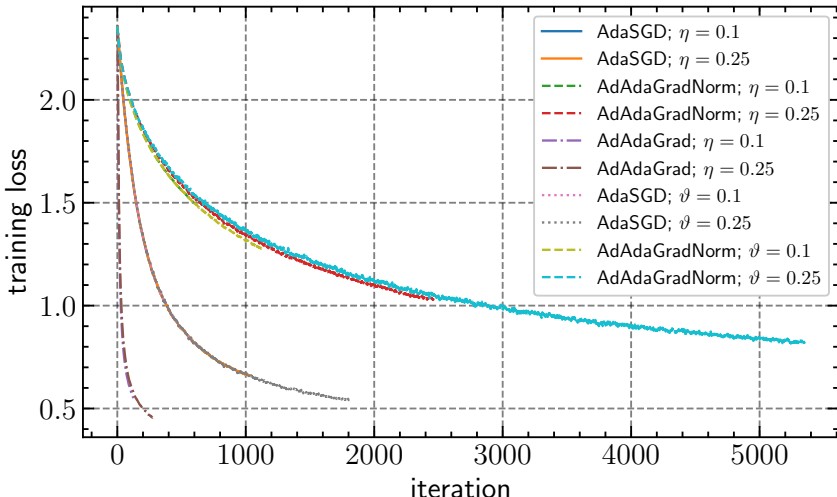

Figure 6: Training loss curves (vs. number of iterations) of ADASGD, ADADAGRAD, and ADADAGRAD-NORM for logistic regression on the MNIST dataset.

## B.2   Three-layer Convolutional Neural Network on MNIST

The following table lists the training specifications and optimizer hyperparameters for the experiments on a three-layer convolutional neural network on the MNIST dataset.

Table 9 provides aggregate cost statistics for the experiments on a three-layer convolutional neural network on the MNIST dataset.

Table 8: Training hyperparameters for three-layer CNN on MNIST

| Model | 3-layer CNN on MNIST |
|---|---|
| Training budget | 6,000,000 samples (100 epochs) |
| Weight initialization | Default |
| Optimizer | SGD or AdaGrad(-Norm) |
| Base learning rate | 0.008 |
| Base batch size | 8 |
| Maximum batch size | 60,000 |
| Weight decay | 0 |
| Momentum | 0 |
| Precision | tf32 |

Table 9: Aggregate cost statistics for three-layer CNN on MNIST

| Scheme | test | time (h) | samples | steps | batch size | loss | accuracy |
|---|---|---|---|---|---|---|---|
| SGD | N/A | 6.77 | 6,000,640 | 2929 | 2048 | 0.12 | 0.96 |
| SGD | N/A | 3.38 | 6,000,640 | 1464 | 4096 | 0.20 | 0.94 |
| SGD | N/A | 1.72 | 6,004,736 | 732 | 8192 | 0.32 | 0.91 |
| SGD | N/A | 0.90 | 6,012,928 | 366 | 16384 | 0.51 | 0.87 |
| SGD | N/A | 0.45 | 6,029,312 | 183 | 32768 | 1.54 | 0.75 |
| SGD | N/A | 0.27 | 6,000,000 | 99 | 60000 | 2.15 | 0.66 |
| AdaGrad | N/A | 7.12 | 6,000,640 | 2929 | 2048 | 0.02 | 0.99 |
| AdaGrad | N/A | 3.60 | 6,000,640 | 1464 | 4096 | 0.02 | 0.99 |
| AdaGrad | N/A | 1.82 | 6,004,736 | 732 | 8192 | 0.05 | 0.98 |
| AdaGrad | N/A | 0.92 | 6,012,928 | 366 | 16384 | 0.07 | 0.98 |
| AdaGrad | N/A | 0.52 | 6,000,000 | 199 | 30000 | 0.10 | 0.97 |
| AdaGrad | N/A | 0.47 | 6,029,312 | 183 | 32768 | 0.11 | 0.97 |
| AdaGrad | N/A | 0.36 | 6,000,000 | 149 | 40000 | 0.13 | 0.96 |
| AdaGrad | N/A | 0.29 | 6,000,000 | 99 | 60000 | 0.17 | 0.95 |
| AdaSGD | $\eta = 0.10$ | 0.73 | 6,051,456 | 256 | 23546 | 0.79 | 0.83 |
| AdaSGD | $\eta = 0.25$ | 1.05 | 6,000,868 | 383 | 15627 | 0.48 | 0.88 |
| AdAdaGrad-Norm | $\eta = 0.10$ | 0.65 | 6,030,808 | 226 | 26567 | 0.88 | 0.83 |
| AdAdaGrad-Norm | $\eta = 0.25$ | 1.27 | 6,030,096 | 435 | 13830 | 0.54 | 0.87 |
| AdAdaGrad | $\eta = 0.10$ | **0.45** | **6,008,592** | **149** | **40057** | **0.15** | **0.96** |
| AdAdaGrad | $\eta = 0.25$ | **0.58** | **6,000,412** | **198** | **30152** | **0.13** | **0.97** |
| AdAdaGrad | $\eta = 0.50$ | **0.62** | **6,035,096** | **215** | **27940** | **0.11** | **0.97** |
| AdAdaGrad | $\eta = 0.75$ | **0.79** | **6,046,180** | **271** | **22228** | **0.10** | **0.97** |
| AdaSGD | $\vartheta = 0.01$ | 0.63 | 6,024,044 | 230 | 26078 | 0.98 | 0.80 |
| AdaSGD | $\vartheta = 0.05$ | 1.17 | 6,047,004 | 411 | 14593 | 0.45 | 0.88 |
| AdAdaGrad-Norm | $\vartheta = 0.01$ | 0.70 | 6,019,232 | 241 | 24872 | 0.83 | 0.84 |
| AdAdaGrad-Norm | $\vartheta = 0.05$ | 1.44 | 6,012,544 | 528 | 11365 | 0.50 | 0.88 |

In addition to the training hyperparameters, we provide additional plots for the norm test and the inner product-based test and briefly compare their batch-size dynamics. From Figure 7, we observe that, for the runs shown, the norm test yields faster and larger batch-size increases than the inner product test, even for small $\vartheta$. In applications where sharp batch-size increases may induce training instability, the more gradual behavior of the inner product test can be preferable.

We also provide two additional sets of plots comparing constant-batch and adaptive-batch variants of SGD and AdaGrad, respectively; see Figures 9 and 11. These plots illustrate how adaptive batch sizes based on the norm test or the inner product-based test compare with a range of constant batch sizes. We observe that the adaptive schemes generally attain validation

accuracies between those obtained with small and large constant batches, suggesting that they can balance computational efficiency and generalization when larger batches are desired.

We also give their corresponding iteration-based plots of the training loss curves in Figures 8, 10 and 12.

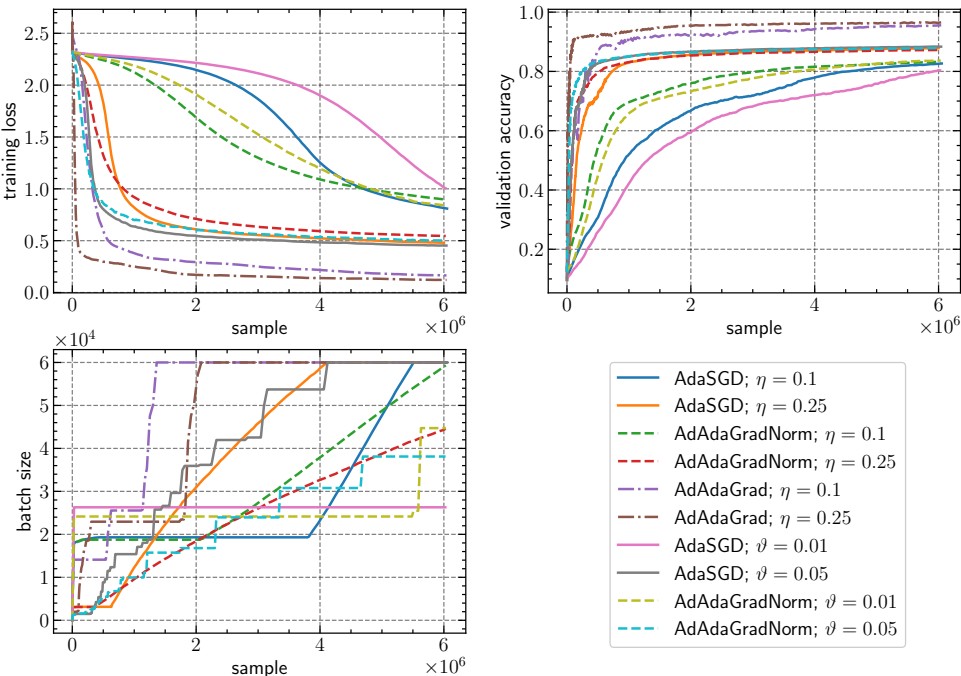

Figure 7: Training loss, validation accuracy and batch size curves (vs. number of training samples) of ADASGD, ADADAGRAD and ADADAGRAD-NORM for three-layer CNN on the MNIST dataset.

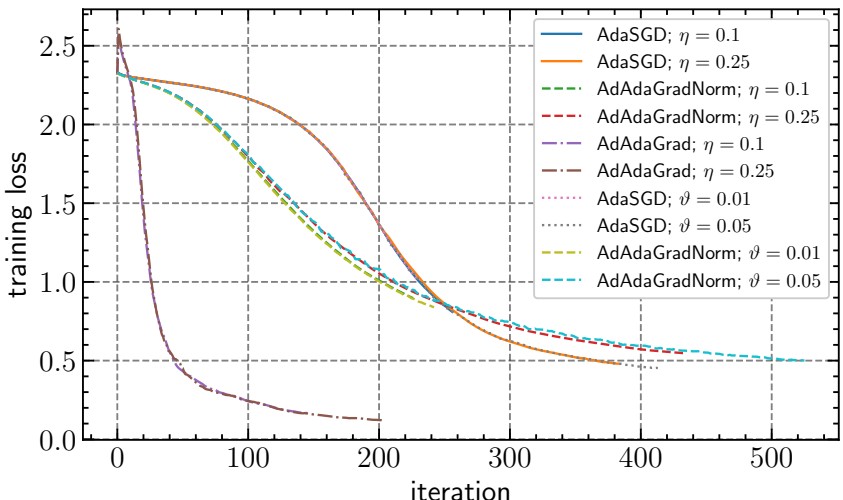

Figure 8: Training loss curves (vs. number of iterations) of ADASGD, ADADAGRAD, and ADADAGRAD-NORM for three-layer CNN on the MNIST dataset.

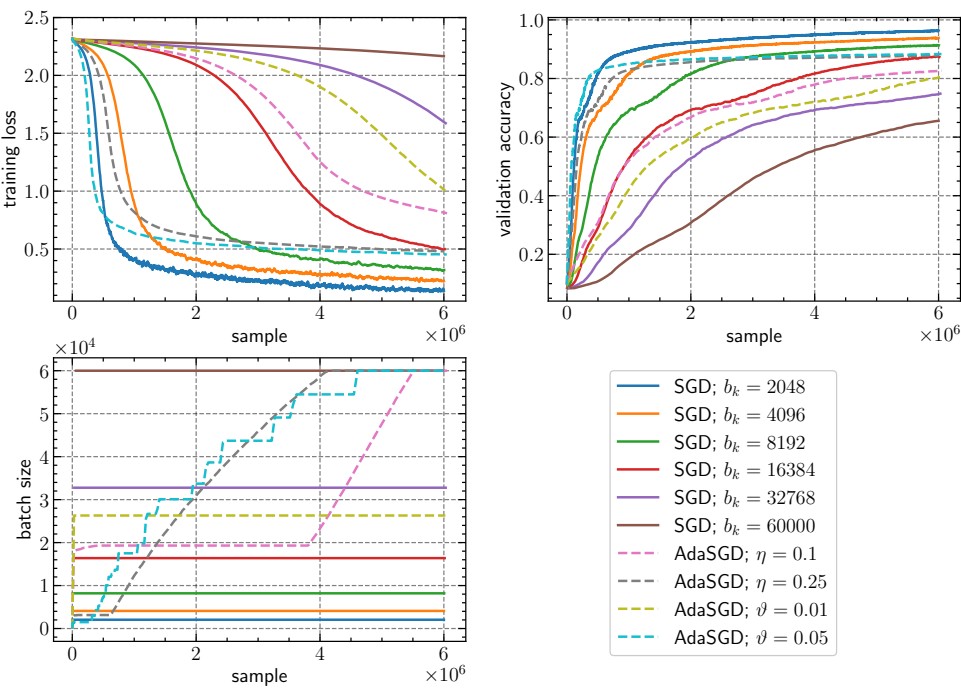

Figure 9: Training loss, validation accuracy and batch size curves (vs. number of training samples) of SGD and ADASGD for three-layer CNN on the MNIST dataset.

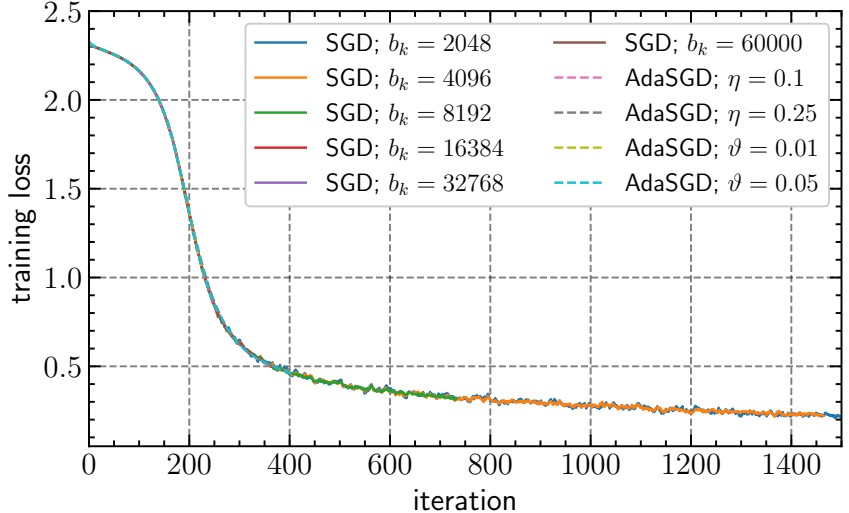

Figure 10: Training loss curves (vs. number of iterations) of SGD and ADASGD for three-layer CNN on the MNIST dataset.

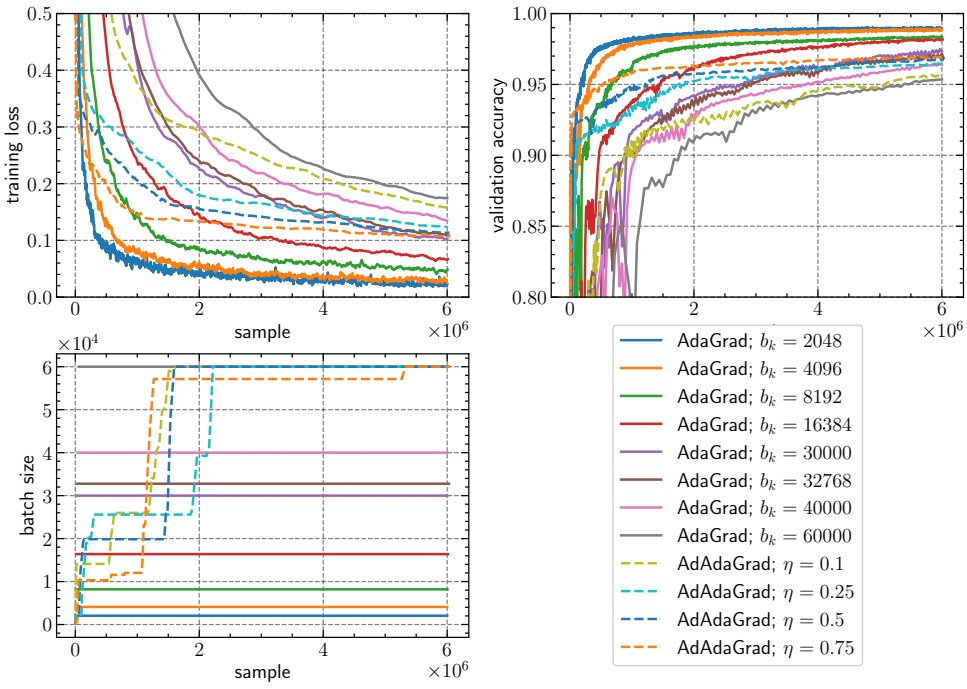

Figure 11: Training loss, validation accuracy and batch size curves (vs. number of training samples) of ADAGRAD and ADADAGRAD for three-layer CNN on the MNIST dataset.

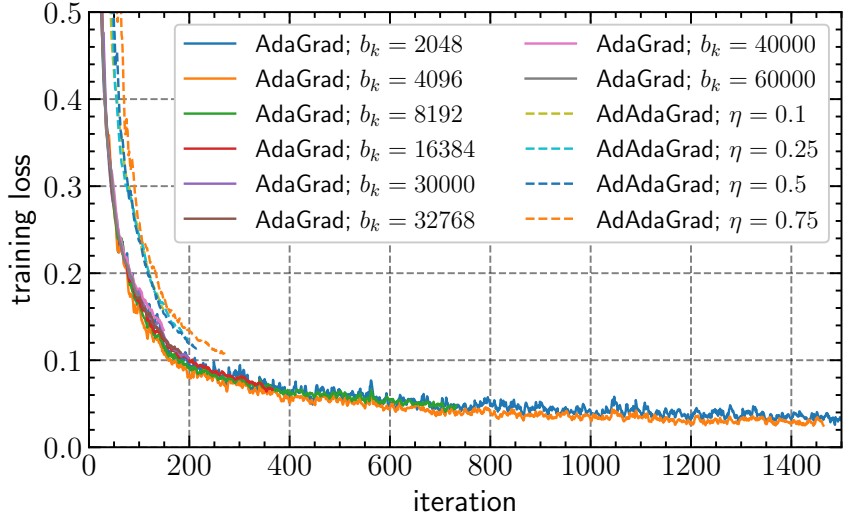

Figure 12: Training loss curves (vs. number of iterations) of ADAGRAD and ADADAGRAD for three-layer CNN on the MNIST dataset.

## B.3 Three-layer Convolutional Neural Network on CIFAR-10

The following table lists the training specifications and optimizer hyperparameters for the experiments on a three-layer convolutional neural network on the CIFAR-10 dataset. As noted in the main text, to reduce the computational overhead of adaptive testing, the test is evaluated every 10 steps in these experiments.

Table 10: Training hyperparameters for three-layer CNN on CIFAR-10

| Model | 3-layer CNN on CIFAR-10 |
|---|---|
| Training budget | 5,000,000 samples (100 epochs) |
| Weight initialization | Default |
| Learning rate schedule | None |
| Optimizer | SGD or ADAGRAD(-NORM) |
| Optimizer scaling rule | None |
| Base learning rate | 0.05 |
| Base batch size | 2 |
| Maximum batch size | 10,000 |
| Weight decay | 0 |
| Momentum | 0 |
| Precision | tf32 |

Table 11 provides aggregate cost statistics for the experiments on a three-layer convolutional neural network on the CIFAR-10 dataset.

Table 11: Aggregate cost statistics for three-layer CNN on CIFAR-10

| Scheme | test | time (h) | samples | steps | batch size | loss | accuracy |
|---|---|---|---|---|---|---|---|
| ADASGD | $\eta = 0.25$ | 0.87 | 5,001,302 | 523 | 9544 | 1.68 | 0.40 |
| ADASGD | $\eta = 0.50$ | 0.96 | 5,003,402 | 658 | 7592 | 1.59 | 0.43 |
| ADADAGRAD-NORM | $\eta = 0.25$ | **0.87** | **5,001,302** | **531** | **9401** | **1.36** | **0.52** |
| ADADAGRAD-NORM | $\eta = 0.50$ | **1.19** | **5,002,835** | **1261** | **3964** | **1.19** | **0.57** |
| ADADAGRAD | $\eta = 0.25$ | 1.05 | 5001581 | 903 | 5533 | 1.20 | 0.54 |
| ADADAGRAD | $\eta = 0.50$ | **1.15** | **5,003,236** | **1123** | **4451** | **1.11** | **0.57** |
| ADASGD | $\vartheta = 0.05$ | **1.38** | 5,001,178 | 1597 | **3130** | **1.17** | **0.57** |
| ADASGD | $\vartheta = 0.10$ | 0.96 | 5,003,342 | 640 | 7806 | 1.64 | 0.41 |
| ADADAGRAD-NORM | $\vartheta = 0.05$ | **0.99** | **5,008,602** | **780** | **6413** | **1.29** | **0.55** |
| ADADAGRAD-NORM | $\vartheta = 0.10$ | 1.54 | 5,003,894 | 1948 | 2567 | 1.13 | 0.58 |

We give the iteration-based plot of the training loss curves in Figure 13.

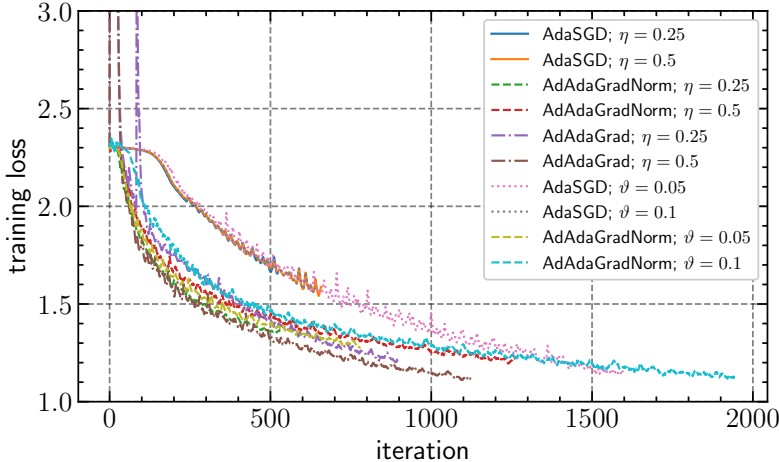

Figure 13: Training loss curves (vs. number of iterations) of ADASGD, ADADAGRAD, and ADADAGRAD-NORM for a three-layer CNN on the CIFAR-10 dataset.

## B.4 RESNET-18 on CIFAR-10

The following table lists the training specifications and optimizer hyperparameters for the experiments on RESNET-18 on the CIFAR-10 dataset.

Table 12: Training hyperparameters for RESNET-18 on CIFAR-10

| Model | RESNET-18 on CIFAR-10 |
|---|---|
| Training budget | 10,000,000 samples (200 epochs) |
| Weight initialization | Default |
| Optimizer | ADAGRAD or ADAM |
| Learning rate schedule | Linear warmup + cosine decay |
| Learning rate warmup (samples) | 1M |
| $(\beta_1, \beta_2)$ | $(0.9, 0.95)$ |
| $\varepsilon$ | $10^{-8}$ |
| Peak learning rate | 0.05 |
| Minimum learning rate | 0.005 |
| Base batch size | 8 |
| Maximum batch size | 50,000 |
| Weight decay | 0 |
| Precision | tf32 |

Table 13 provides aggregate cost statistics for the experiments on RESNET-18 on the CIFAR-10 dataset, while iteration-based plots of the training loss curves are given in Figures 14 and 15.

Table 13: Aggregate cost statistics for RESNET-18 on CIFAR-10

| Scheme | test | time (h) | samples | steps | batch size | loss | accuracy |
|---|---|---|---|---|---|---|---|
| ADAGRAD | N/A | 1.21 | 10,002,432 | 2441 | 4096 | 0.0042 | 0.8521 |
| ADAGRAD | N/A | 0.97 | 10,002,432 | 1220 | 8192 | 0.0808 | 0.8072 |
| ADAGRAD | N/A | 0.77 | 10,010,624 | 610 | 16384 | 0.5098 | 0.7264 |
| ADAGRAD | N/A | 0.45 | 10,027,008 | 305 | 32768 | 0.9684 | 0.5816 |
| ADAGRAD | N/A | 0.33 | 10,000,000 | 199 | 50000 | 1.3625 | 0.4708 |
| ADAM | N/A | 1.20 | 10,002,432 | 2441 | 4096 | 0.0003 | 0.9147 |
| ADAM | N/A | 0.97 | 10,002,432 | 1220 | 8192 | 0.0004 | 0.8946 |
| ADAM | N/A | 0.77 | 10,010,624 | 610 | 16384 | 0.0028 | 0.8628 |
| ADAM | N/A | 0.45 | 10,027,008 | 305 | 32768 | 0.4000 | 0.7463 |
| ADAM | N/A | 0.33 | 10,000,000 | 199 | 50000 | 1.0680 | 0.5750 |
| ADADAGRAD | $\eta = 0.025$ | 0.32 | 10,020,200 | 222 | 44934 | 1.2770 | 0.5107 |
| ADADAGRAD | $\eta = 0.050$ | 0.60 | 10,018,824 | 485 | 20615 | 0.6204 | 0.7079 |
| ADADAGRAD | $\eta = 0.075$ | **1.02** | **10,004,880** | **1697** | **5892** | **0.0258** | **0.8180** |
| ADADAGRAD | $\eta = 0.100$ | **0.94** | **10,008,128** | **1404** | **7123** | **0.0668** | **0.8085** |
| ADADAM | $\eta = 0.025$ | 0.34 | 10,044,552 | 211 | 47380 | 0.9039 | 0.6234 |
| ADADAM | $\eta = 0.050$ | 0.60 | 10,018,824 | 426 | 23463 | 0.0061 | 0.8228 |
| ADADAM | $\eta = 0.075$ | 0.74 | 10,008,248 | 900 | 11108 | 0.0008 | 0.8983 |
| ADADAM | $\eta = 0.100$ | **0.82** | **10,007,788** | **1126** | **8880** | **0.0000** | **0.9042** |

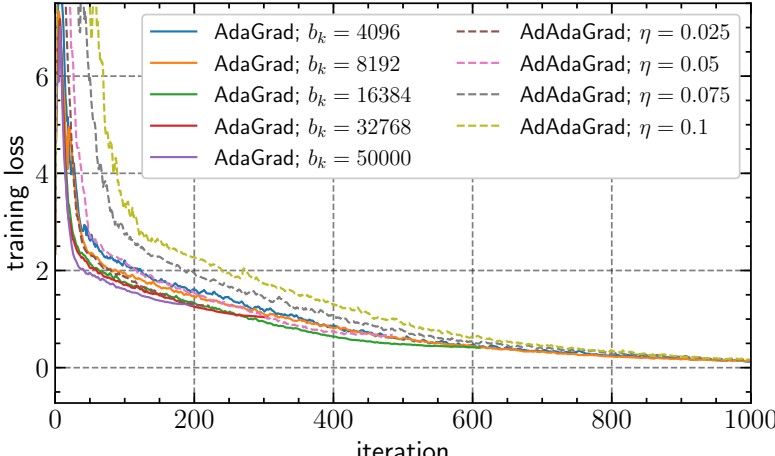

Figure 14: Training loss curves (vs. number of iterations) of ADAGRAD and ADADAGRAD for RESNET-18 on the CIFAR-10 dataset.

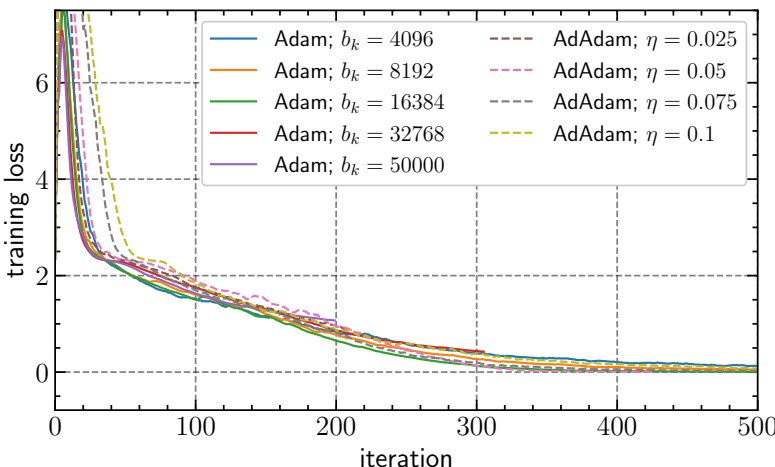

Figure 15: Training loss curves (vs. number of iterations) of ADAM and ADADAM for RESNET-18 on the CIFAR-10 dataset.

## Code Availability

All analysis code is available in the GitHub repository: [removed in the anonymized version]

