# OpenReview forum: "AdAdaGrad: Adaptive Batch Size Schemes for Adaptive Gradient Methods"
_SLADS/Section_C — Decision pending for SLADS_Section_C_

### Review · Reviewer_YZA3 · 2026-06-08

**Summary Of Contributions:**

This paper proposes two adaptive gradient methods, AdAdaGrad and AdAdaGrad-Norm, which combine adaptive batch-size schemes with AdaGrad-type updates to balance training efficiency and generalization in large-batch training. Building on the adaptive sampling framework of Bollapragada et al. (2018a), the authors extend such ideas to adaptive gradient methods and establish high-probability sublinear convergence guarantees for smooth nonconvex objectives. The analysis is further extended to objectives satisfying generalized smoothness conditions. Numerical experiments on image-classification tasks show that the proposed schemes can improve training efficiency and generalization performance compared with several baseline methods.

**Audience:**

Yes

**Broader Impact Concerns:**

I do not see any significant broader impact concerns.

**Claims And Evidence:**

Yes

**Requested Changes:**

1. It would be helpful to include a table comparing the convergence rates of the proposed methods with those of existing stochastic gradient methods, including SGD with adaptive sampling, AdaGrad, and AdaGrad-Norm. In particular, please clarify whether the proposed methods achieve the optimal convergence rate under the considered assumptions.

2. On Page 6, the notation $\mathfrak{F}_k$ appears to denote a filtration or $\sigma$-algebra. Please provide a precise definition.

3. Bollapragada et al. (2018a) have already established convergence results for adaptive sampling methods combined with SGD. In my view, the main difference of this paper is that it considers adaptive sampling for adaptive gradient methods, especially AdaGrad and AdaGrad-Norm. Therefore, the authors should clearly explain the new technical challenges in proving convergence for these algorithms. It would also be useful to highlight what new and interesting insights are obtained from the convergence analysis beyond the SGD setting.

4. In Theorem 6, does the stepsize parameter $\alpha$ need to satisfy an upper bound? Please provide a clearer comment on this theorem. In addition, the bound involves the parameter $\rho$. Could the authors discuss how to choose $\rho$ in practice, and whether it makes the bound tighter?

5. In the numerical experiments, the batch sizes used are quite large. In practice, batch sizes such as 128 or 256 are commonly used in deep learning. It would be useful to compare the proposed methods with standard SGD or adaptive gradient methods under such traditional batch-size settings. Moreover, please include plots of training loss versus the number of iterations, in addition to training loss versus the number of processed samples. This would make the empirical results more consistent with the iteration-based convergence guarantees in the theory.

**Strengths And Weaknesses:**

Strengths. A main strength of the paper lies in its theoretical analysis. The authors establish high-probability sublinear convergence guarantees for smooth nonconvex objectives and further extend the analysis to the generalized smoothness setting.

Weaknesses. The experimental evaluation is relatively limited in scale. The experiments mainly focus on standard image-classification benchmarks and relatively small models, so it remains unclear how the proposed schemes would perform for truly large-scale models, such as transformers.

---

> ### Author Response · Authors · 2026-06-26
> **Response**
>
> We thank the reviewer for the positive assessment of the theoretical contribution and for the helpful suggestions. We will add a convergence-rate comparison table contrasting the proposed methods with SGD with adaptive sampling, standard SGD, AdaGrad, and AdaGrad-Norm under the relevant assumptions. We will also clarify that the faster $\mathscr{O}(1/K)$ stationarity rate is obtained under exact adaptive-sampling conditions, which imply an expected strong-growth-type condition along the iterates, and therefore should not be interpreted as an unconditional improvement over the usual stochastic nonconvex rates under bounded variance.
>
> We will also expand the discussion of the technical challenges beyond the SGD setting. The key new issue is that in AdaGrad-Norm and AdaGrad the stochastic gradient appears both in the numerator of the update and in the adaptive denominator. Hence, even when $g_k$ is conditionally unbiased, the normalized direction $g_k/\sqrt{v_k}$ is not handled by the same argument as SGD because $v_k$ depends on $g_k$. Our proof controls this coupling through a decomposition relative to $v_{k-1}$, a potential-difference argument, and logarithmic AdaGrad-type summability bounds. For coordinate-wise AdaGrad, the same issue occurs coordinate by coordinate, which explains the need for a coordinate-wise variance-control condition in the theorem.
>
> We will make the notation for the filtration precise. Specifically, we will define ${\cal F}_k \coloneqq\sigma(\lbrace x_1, {\cal B}\_1,\ldots, {\cal B}\_{k-1}\rbrace)$ as the information available before drawing ${\cal B}_k$, and write $\\mathbb{E}_k[\cdot]\\coloneqq\\mathbb{E}[\cdot\mid\mathcal{F}_k]$.
>
> Regarding Theorem 6, we will add a comment that under $L$-Lipschitz smoothness the theorem allows any constant $\alpha>0$; no upper bound on $\alpha$ is required for the statement. Nevertheless, the constants depend on $\alpha$, so very large values may lead to loose bounds in practice. We will also explain that $\rho>1$ is a proof parameter rather than an algorithmic hyperparameter; fixing $\rho=2$ is sufficient for interpreting the result, while optimizing it can only improve constants and does not change the asymptotic rate.
>
> Finally, we will clarify the empirical comparison with constant-batch-size methods. The current experimental suite already includes constant-batch-size baselines for the CNN on MNIST and ResNet-18 on CIFAR-10, with batch sizes chosen to cover the practically relevant large-batch regimes reached by the adaptive methods. We agree that comparisons with nonadaptive batch-size choices are important, and in the revision we will make these comparisons more prominent, include them in the reproducibility and aggregate-cost tables, and add training-loss plots versus optimizer iterations in addition to the existing plots versus processed samples. We do not focus on very small constant batch sizes such as 128 or 256, because the purpose of the adaptive tests is to determine when the method can safely move toward larger batches; thus, the most informative nonadaptive alternatives are constant batch sizes in the same large-batch regime that the adaptive methods are designed to reach. This revised presentation allows the reader to compare adaptive and constant-batch-size methods both in terms of processed samples and in terms of optimizer iterations.

---

> > ### Comment · Reviewer_YZA3 · 2026-06-29
> > **Reply of the response**
> >
> > I thank the authors for their elaborations. Their original response has addressed my concern on that aspect. Could you provide the meaning of $\delta$ in Table 1?

---

> > > ### Author Response · Authors · 2026-06-29
> > >
> > > Thank you. We added the meaning of $\delta$ in the caption of Table 1 in the revised pdf.

---

### Review · Reviewer_7wai · 2026-06-17

**Summary Of Contributions:**

The paper proposes adaptive batch-size schemes for adaptive gradient methods, namely AdAdaGrad and AdAdaGrad-Norm, by incorporating adaptive sampling tests into AdaGrad-type training so as to balance the computational efficiency of large-batch methods with the generalization behavior often associated with smaller batches. It proves high-probability convergence guarantees for AdAdaGrad-Norm on smooth nonconvex objectives, extends the analysis to a coordinate-wise adaptive sampling variant for AdAdaGrad, and supports the proposed schemes with image-classification experiments demonstrating favorable trade-offs between training efficiency and generalization.

**Audience:**

Yes

**Broader Impact Concerns:**

None.

**Claims And Evidence:**

Yes

**Requested Changes:**

C1. Provide convergence guarantees for the implemented empirical batch-size rule, or clearly restrict the theoretical claim.

The current convergence analysis is based on exact adaptive sampling tests, while the implemented algorithm uses empirical batch statistics to choose batch sizes. Moreover, the practical rule uses the current batch gradient to perform the update, while the test statistic computed from the current batch determines a future batch size. Therefore, the theorem does not yet directly apply to the empirical algorithm used in the experiments.

To make the theoretical and empirical parts consistent, the authors should either provide a convergence analysis under conditions that match the implemented empirical batch-size rule, or modify the algorithm so that the batch used for the current update is guaranteed to satisfy the required test. For example, one possible route would be to impose suitable concentration assumptions on per-sample gradients and show that the empirical norm test implies the exact variance condition with high probability. If such an analysis is not included, the paper should explicitly state that the convergence theorem applies to an idealized exact-test scheme rather than to the fully implemented empirical algorithm.

C2. Provide theory for the AdAdaGrad variant actually used in the experiments, or clearly separate it from the analyzed method.

For coordinate-wise AdaGrad, the stated convergence theorem relies on a coordinate-wise exact norm test. However, the experiments use the standard norm test rather than the coordinate-wise test, because the latter is computationally more expensive. If this is the empirical version the authors intend to advocate, then the paper should provide a convergence guarantee for AdAdaGrad under the standard norm test, possibly with additional assumptions that are consistent with the experimental implementation.

Alternatively, the authors may add experiments using the coordinate-wise test analyzed in the theorem. If neither is done, the paper should clearly state that the main AdAdaGrad experiments are not directly covered by the current convergence theory.

C3. Improve computational-cost reporting.

As discussed in W3, the adaptive tests require additional computation beyond standard minibatch gradients. The experiments should report wall-clock time consistently, total number of processed samples, test frequency, memory overhead, and the overhead of computing per-sample gradient statistics. This would make the comparison with fixed-batch AdaGrad/AdaGrad-Norm and related baselines more informative.

C4. Add implementation and reproducibility details.

As discussed in W4, the paper should specify the practical batch-size caps, safeguards for small denominators, the sampling protocol, and the exact test parameters and test frequencies used in each experiment. If the augmented inner product test is used, both parameters $\vartheta$ and $\nu$ should be reported.

C5. Moderate the interpretation of the theoretical guarantees.

The paper should avoid implying that the theory fully explains every empirical variant. A clearer separation between the exact-test method analyzed in the theorems and the empirical approximate-test implementation would make the contribution more precise and more convincing.

**Strengths And Weaknesses:**

Strengths

S1. The interaction between batch size and stepsize is a central issue in stochastic optimization, especially for large-scale machine learning. While adaptive gradient methods are widely used in deep learning, principled batch-size adaptation for such methods is comparatively less studied. Thus, the topic of the paper is practically relevant and timely.

S2. The paper builds on the norm test and augmented inner product test from the adaptive sampling literature. This connection is appropriate and gives the proposed algorithms a more principled basis than purely heuristic progressive batching rules.

S3. The experiments cover convex logistic regression, nonconvex CNN training, and ResNet-18 on CIFAR-10. The paper also reports batch-size trajectories, which are important for assessing whether the proposed tests behave as intended.

Weaknesses

W1. The theoretical guarantees should be better aligned with the implemented empirical batch-size rule.
The convergence theorems are proved under exact adaptive sampling tests, such as
$$
\mathbb{E} |g_k-\nabla F(x_k)|^2
\leq
\eta^2 |\nabla F(x_k)|^2.
$$
This is a reasonable theoretical condition. However, the practical algorithm uses empirical batch statistics to approximate these tests. More importantly, the implemented rule appears to use the current batch gradient to perform the update, while the test statistic computed from the current batch is used to determine a future batch size. Thus, the current batch used for the update is not necessarily guaranteed to satisfy the exact test required by the theorem.

Therefore, the current theory is best interpreted as a guarantee for an idealized exact-test batch-size scheme, rather than a complete guarantee for the fully implemented empirical rule. This is not necessarily a flaw in the theoretical analysis, but it limits the extent to which the theory explains the empirical algorithm. To make the theory and experiments consistent, the authors should either provide convergence guarantees under assumptions that match the implemented empirical batch-size rule, or explicitly state that the convergence theory applies to an idealized exact-test version of the method.

W2. The theoretical result for AdAdaGrad appears narrower than the empirical implementation.
For coordinate-wise AdaGrad, the stated theorem relies on a coordinate-wise exact norm test,
$$
\mathbb{E} \left[
\left(\nabla_j F_{B_k}(x_k)-\nabla_j F(x_k)\right)^2
\right]
\leq
\eta^2\left(\nabla_j F(x_k)\right)^2,
\qquad j=1,\ldots,d.
$$
However, the experiments use the standard norm test rather than the coordinate-wise test, because the latter is computationally expensive in high dimensions. Hence, the main AdAdaGrad experiments are not directly covered by the coordinate-wise AdaGrad theorem.

Since AdAdaGrad is one of the main proposed methods, this mismatch should be addressed. The authors should either provide a convergence analysis for the standard-norm-test version of AdAdaGrad used in the experiments, add experiments using the coordinate-wise test analyzed in the theorem, or clearly separate the empirically implemented AdAdaGrad variant from the theoretically analyzed one.

W3. The computational cost of the adaptive tests should be reported more systematically.
The proposed tests require estimating gradient variance or related per-sample gradient quantities. This can introduce non-negligible overhead compared with standard minibatch training. The paper reports wall-clock time in some experiments and describes the use of torch.func/vmap for per-sample gradients, which is useful. However, the cost accounting is not fully systematic across all experiments, and the overhead of computing the adaptive test statistics is not isolated.

Since the paper is about batch-size adaptation, it would be helpful to report, consistently across experiments, wall-clock time, total processed samples, test frequency, memory overhead, and the cost of computing the required per-sample quantities. This would make the comparison with fixed-batch AdaGrad/AdaGrad-Norm and related baselines more informative.

W4. Some implementation details should be made more explicit and systematic.
The paper provides several implementation details, such as maximum batch sizes in the experimental settings and a limiting interpretation when denominators in the test statistics vanish. However, the practical safeguards used in the actual implementation are not always fully specified. For example, it would be useful to clarify how very small empirical gradient norms are handled numerically, whether sampling is performed with replacement, without replacement, or through shuffled epochs, and how frequently the adaptive tests are evaluated in each experiment. These details are important for reproducibility and for understanding how the adaptive tests behave near stationary points.

---

> ### Author Response · Authors · 2026-06-26
> **Response**
>
> We thank the reviewer for the careful and constructive comments. We agree that the current draft should more clearly distinguish the exact-test algorithms analyzed in the convergence theory from the empirical approximate-test rules used in implementation. In the revision, we will make this distinction explicit in the introduction and in the statements surrounding Theorems 6--9. In particular, we will state that the current convergence guarantees apply to an idealized exact-test version of AdAdaGrad-Norm and AdAdaGrad, where the required variance conditions hold at the iterate used for the update. The empirical implementation uses batch-based estimates of these quantities and uses the current batch statistics to choose the next batch size; hence, the theory should be interpreted as justifying the population-level adaptive-sampling principle rather than as a full finite-sample guarantee for every implementation detail.
>
> We also agree with the reviewer that the current AdAdaGrad theorem and the main AdAdaGrad experiments should be separated more clearly. The theorem for coordinate-wise AdaGrad uses a coordinate-wise exact norm test, whereas the practical experiments use the standard norm test because the coordinate-wise test is expensive in high-dimensional neural networks. In the revision, we will explicitly identify this as a limitation of the present theory. We will revise the text so that the theoretically analyzed method is described as coordinate-wise exact-test AdAdaGrad, while the empirical method is described as the practical standard-norm-test AdAdaGrad variant. We will avoid implying that Theorem 8 directly covers the experimental AdAdaGrad variant.
>
> We will add a dedicated implementation/reproducibility paragraph clarifying the sampling protocol, test frequency, batch-size caps, and numerical safeguards. Specifically, all reported experiments use the standard shuffled-epoch protocol: the data are shuffled at the beginning of each epoch and minibatches are processed sequentially from this shuffled order, rather than sampled independently with replacement. In the adaptive-batch experiments, the batch-size statistic computed from the current minibatch is used to determine the batch size for the next adaptive update. The adaptive test is evaluated once per optimizer iteration during the adaptive phase, except for the three-layer CNN on CIFAR-10 experiment, where it is evaluated every $10$ optimizer iterations. Once the prescribed maximum batch size is reached, the adaptive test is no longer evaluated and training proceeds with the capped batch size. We will also clarify that no denominator clipping or special small-gradient safeguard was used in the reported runs. Empirically, the maximum batch size was typically reached before the very small-gradient regime, so small empirical gradient norms in the test denominators did not create numerical issues in these experiments.
>
> We further agree that the computational cost of the adaptive tests should be reported more systematically. In the revision, we will add reproducibility and aggregate-cost tables for the experimental runs, including wall-clock time, total number of processed samples, number of optimizer iterations, average batch size, final training loss, and final validation accuracy. These tables will make the comparison between adaptive-batch methods and constant-batch-size baselines more transparent. The reported wall-clock times include the cost of computing the empirical adaptive-test statistics, including the per-sample gradient quantities used by the tests. However, our current implementation records aggregate end-to-end wall-clock time rather than separately instrumenting the forward/backward pass, the per-sample-gradient-statistic computation, the batch-size-test computation, or GPU memory usage. We will state this limitation explicitly and describe the reported timings as aggregate cost measurements, rather than as an isolated decomposition of adaptive-test overhead or memory overhead.
>
> Finally, we have added additional plots of training loss versus the number of optimizer iterations, complementing the original plots versus the number of processed samples. The iteration-based plots connect the experiments more directly to the iteration-complexity form of the theoretical guarantees, while the sample-based plots continue to illustrate the sample-efficiency behavior of adaptive batch sizing.

---

> > ### Comment · Reviewer_7wai · 2026-07-17
> >
> > Thanks for the response and the effort in addressing my comments. I have no further comments and have submitted my recommendation.

---

### Author Response · Authors · 2026-06-26
**Revised Submission PDF**

We have revised our submission PDF to address the requested changes of both reviewers. In particular, we added clarifications of the theoretical contributions of our work and further experimental details (including tables and additional plots) in both the main text and the appendix. All new/revised materials are in navy blue color.

---

### Decision · Action_Editor_koGv · 2026-07-19

**Recommendation:** Accept as is

**Audience:**

Yes

**Claims And Evidence:**

Yes